# Role of SAGA in the asymmetric segregation of DNA circles during yeast ageing

**Annina Denoth-Lippuner, Marek Konrad Krzyzanowski, Catherine Stober, Yves Barral***

Institute of Biochemistry, Department of Biology, ETH Zürich, Zürich, Switzerland

**Abstract** In eukaryotes, intra-chromosomal recombination generates DNA circles, but little is known about how cells react to them. In yeast, partitioning of such circles to the mother cell at mitosis ensures their loss from the population but promotes replicative ageing. Nevertheless, the mechanisms of partitioning are debated. In this study, we show that the SAGA complex mediates the interaction of non-chromosomal DNA circles with nuclear pore complexes (NPCs) and thereby promotes their confinement in the mother cell. Reciprocally, this causes retention and accumulation of NPCs, which affects the organization of ageing nuclei. Thus, SAGA prevents the spreading of DNA circles by linking them to NPCs, but unavoidably causes accumulation of circles and NPCs in the mother cell, and thereby promotes ageing. Together, our data provide a unifying model for the asymmetric segregation of DNA circles and how age affects nuclear organization.

## Introduction

Homologous recombination plays an important role in DNA repair in all organisms and acts through the formation of a Holliday junction between the damaged molecule and its repair template. When recombination involves two homologous chromosomes, the resolution of the Holliday junction leads to a cross-over, the exchange of arms between two chromosomes, in 50% of the cases (**Heyer, 2004**). However, when recombination occurs between two loci on the same chromosome, i.e. if the break affects a repeated region of the genome, recombination is intra-chromosomal. In 50% of these events, the resolution of the Holliday junction leads to the excision of a DNA circle containing one or several of the involved repeats (**Gaubatz, 1990**). Accordingly, due to the highly repeated nature of the ribosomal DNA (rDNA) locus, rDNA circles have been found in all eukaryotes tested so far (**Cohen et al., 2010**). Similarly, double-minute, satellite- and t-circles, as well as a multitude of micro DNAs have been identified in animal cells (**Cohen and Segal, 2009**; **Shibata et al., 2012**). However, how cells react to these molecules is not understood. Particularly, it is unclear whether these cells recognize, sort, and eliminate these molecules or let them passively disappear by dilution.

In *Saccharomyces cerevisiae*, extrachromosomal rDNA circles (ERCs) form spontaneously but fail to accumulate in the population, owing to their highly asymmetric segregation upon cell division (**Sinclair and Guarente, 1997**). Indeed, the asymmetry of the budding process generates a mother and a daughter cell with distinct sizes and fate. While most daughter cells are born young, i.e. with a full replicative potential, mother cells age with each budding cycle and therefore generate only a limited number of daughter cells ((**Mortimer and Johnston, 1959**) and reviewed in (**Denoth Lippuner et al., 2014**)). Since rDNA circles contain a replication origin they duplicate in S-phase, and due to the asymmetric segregation they accumulate with time in the mother cell. Several arguments indicate that they contribute to replicative ageing of these mother cells. Artificially introducing an ERC into the bud reduces its life span (**Sinclair and Guarente, 1997**). Likewise, mutations that enhance ERC formation shorten the longevity of mother cells (**Kaeberlein et al., 1999**). Conversely, mutant cells with reduced

*For correspondence: yves.barral@bc.biol.ethz.ch

Competing interests: The authors declare that no competing interests exist.

**eLife digest** Budding yeast is a microorganism that has been widely studied to understand how it and many other organisms, including animals, age over time. This yeast is so named because it proliferates by 'budding' daughter cells out of the surface of a mother cell. For each daughter cell that buds, the mother cell loses some fitness and eventually dies after a certain number of budding events. This process is called 'replicative ageing', and it also resembles the way that stem cells age. In contrast, the newly formed daughters essentially have their age 'reset to zero' and grow until they turn into mother cells themselves.

Several molecules or factors have been linked to replicative ageing. These are retained in the mother cell during budding, rather than being passed on to the daughters. Non-chromosomal DNA circles, for example, are rings of DNA that detach from chromosomes during DNA repair and that accumulate inside the ageing mother cell over time. How the mother cells retain these circles of DNA is an on-going topic of debate.

Similar to plants and animals, chromosomes in yeast cells are confined in a membrane-bound structure known as the cell nucleus. The nuclear membrane is perforated by channels called nuclear pore complexes that ensure the transport of molecules into, and out of, the nucleus. Now, Denoth-Lippuner et al. establish that for the non-chromosomal DNA circles to be efficiently confined in the mother cell, the DNA circles must be anchored to the nuclear pore complexes.

Denoth-Lippuner et al. next asked how the DNA circles were anchored to these complexes; and found that another complex of proteins known as SAGA is involved. When components of the SAGA complex were deleted in budding yeast cells, non-chromosomal DNA circles spread into the daughters as well. On the other hand, artificially anchoring these DNA circles to the nuclear pore complex alleviated the need for the SAGA complex, in order to retain these molecules in the mother cell.

Denoth-Lippuner et al. also show that SAGA-dependent attachment of the DNA circles to the nuclear pore complexes causes these complexes to remain in the mother cell. As a consequence, these nuclear pore complexes accumulate in the mother cells as they age. The number of nuclear pore complexes in the daughter cells, however, remained fairly constant. Together these data raise the question of whether the effects of DNA circles on the number and activity of the nuclear pores might account for their contribution to ageing, perhaps by affecting the workings of the nucleus.

rates of ERC formation, such as *fob1Δ*, or those defective in ERC retention, such as *bud6Δ* mutant cells, live longer (*Defossez et al., 1999*; *Shcheprova et al., 2008*). Interestingly, any artificial circle that replicates in vivo but lacks a partitioning sequence (e.g. a centromere) segregates similarly and promotes replicative ageing when introduced into yeast (*Falcon and Aris, 2003*). Thus, non-chromosomal DNA circles segregate asymmetrically, affect cellular physiology, and promote ageing in a manner that is independent of their DNA sequence. How these DNA circles contribute to ageing is not known.

One approach to characterize the effects of DNA circles on cellular physiology is to study the mechanism of their segregation. Two models have been proposed to explain the retention of circular DNA molecules in the mother cells (*Ouellet and Barral, 2012*). The morpho-kinetic model proposes that circles freely diffuse in the nucleus and that their retention results from the morphology of the dividing yeast nucleus and the short duration of anaphase, which together limit the probability that DNA circles diffuse into the bud (*Gehlen et al., 2011*). Using measured parameters for nuclear geometry and division speed, this model predicts a retention frequency of 0.75–0.90 per individual plasmid. However, mathematical modeling indicates that observed ageing curves require retention frequencies above 0.99 per individual ERC (*Gillespie et al., 2004*), which is higher than what the morpho-kinetic model can achieve. A second model, the barrier model, is based on the observation that a lateral diffusion barrier in the outer membrane of the nuclear envelope impedes the diffusion of membrane proteins through the bud neck and hence their exchange between mother and bud parts of the nucleus (*Shcheprova et al., 2008*; *Boettcher et al., 2012*; *Clay et al., 2014*). This model proposes that DNA circles attach to a receptor in the nuclear envelope to ensure their subsequent

confinement into the mother cell by the lateral diffusion barrier (*Shcheprova et al., 2008*; *Clay et al., 2014*).

The main difference between these models is whether confinement of the circle within the mother cell is purely passive or relies on mechanisms that are able to distinguish non-chromosomal DNA circles from bona-fide chromosomes to promote their specific anchorage and asymmetric segregation. However, no such mechanism is known yet. Whether DNA circles passively diffuse or are recognized by the cell would be predicted to have distinct consequences on the localization of the circles and their effects on nuclear organization. A passive model predicts that DNA circles do not interact specifically with any nuclear structure. Therefore, their accumulation should have little impact on nuclear organization. On the other hand, if cells recognize DNA circles, accumulating circles would increasingly interact with the corresponding structure and should progressively affect its size and organization. Thus, in order to better understand whether and how DNA circles are recognized by the cell, and to shed light on how they interfere with cellular physiology, we investigated how accumulating DNA circles localize and whether they affect nuclear organization.

## Results

### Accumulation of non-centromeric DNA circles leads to the formation of an NPC cap

To investigate the localization and effects of non-centromeric DNA circles on nuclear organization, we used the plasmid pPCM14 (*Figure 1A*), containing a replication origin (ARS1) and 224 repeats of the TetO sequence (*Megee and Koshland, 1999*). In cells expressing a TetR-mCherry fusion protein, which binds the TetO sequence, the plasmid is observed as a focus of red fluorescence. Additionally, the plasmid contains an excisable centromere, leading to the formation of a labeled non-centromeric DNA circle upon expression of the R-recombinase. The plasmid also contains two auxotrophic selection markers: *URA3* located between the two recombination-sites to select against accidental centromere excision and *LEU2* on the residual backbone, allowing selection for the plasmid after centromere excision. In budding yeast, all centromeres co-localize with spindle pole bodies (SPBs) throughout the cell cycle (*Goshima and Yanagida, 2000*). Accordingly, we observe the centromeric plasmid in close proximity to the SPBs (*Figure 1A*). 3 hr after addition of estradiol to induce centromere excision, plasmids are localized away from the SPBs in 68% of the cells. Most of those cells displayed one or two small plasmid foci (61% or 33%, respectively; *Figure 1A*) and only 6% of the cells showed more than two foci. These foci were on average 2.5 times more intense than the nucleoplasmic background (*Figure 1B*) and localized to the vicinity of the nuclear rim (see below, and Figure 5).

In contrast, when the cells were grown in selective media for 16–18 hr, allowing the circles to accumulate through multiple rounds of replication and division, the majority of the cells (93%) contained no plasmid, consistent with the plasmids segregating asymmetrically and accumulating in only few retaining cells. A small fraction of the cells (1%) contained few plasmids visualized as small foci as above, whereas the rest of the cells (6%) carried one large and intense signal, 8-fold above background (*Figure 1B*). Consistent with the idea that this fluorescent mass corresponded to the accumulated plasmids, these cells were the few initial mother cells that had already proceeded through several division cycles, as determined by the visualization of bud scars with calcofluor white (5.2 ± 1.9 bud scars on cells with fluorescent mass compared to 0.7 ± 0.9 bud scars in cells without plasmids). After five rounds of duplication, cells contain approximately 50–100 plasmids (*Hyman et al., 1982*). Remarkably, these accumulated plasmids did not spread throughout the nucleoplasm but rather were collected to a single place at the nuclear periphery.

In order to characterize this structure in more detail, the accumulated plasmids were visualized in wild-type cells co-expressing TetR-mCherry with different nuclear markers fused to GFP. The nucleoporin Nup82 tagged with three copies of a super-folder GFP (Nup82-3sfGFP) was used to visualize the morphology of the nuclear envelope; Spc42-GFP indicated the position of the SPB and Fob1-GFP labeled the nucleolus. In most cases, the accumulated DNA circles localized adjacent to, but were excluded from, the nucleolus (Fob1-GFP; *Figure 1C*). Accordingly, they localized opposite to the SPB. Labeling of the nuclear pore complexes (NPCs) indicated that the nuclei loaded with DNA circles were larger than those with no or few circles (average volume increased 1.7-fold; N = 50 cells). Remarkably, accumulation of DNA circles drastically affected the NPC distribution. The circle-loaded cells contained on an average 2.7-fold more nuclear pores compared to cells without plasmids (*Figure 1C*). This

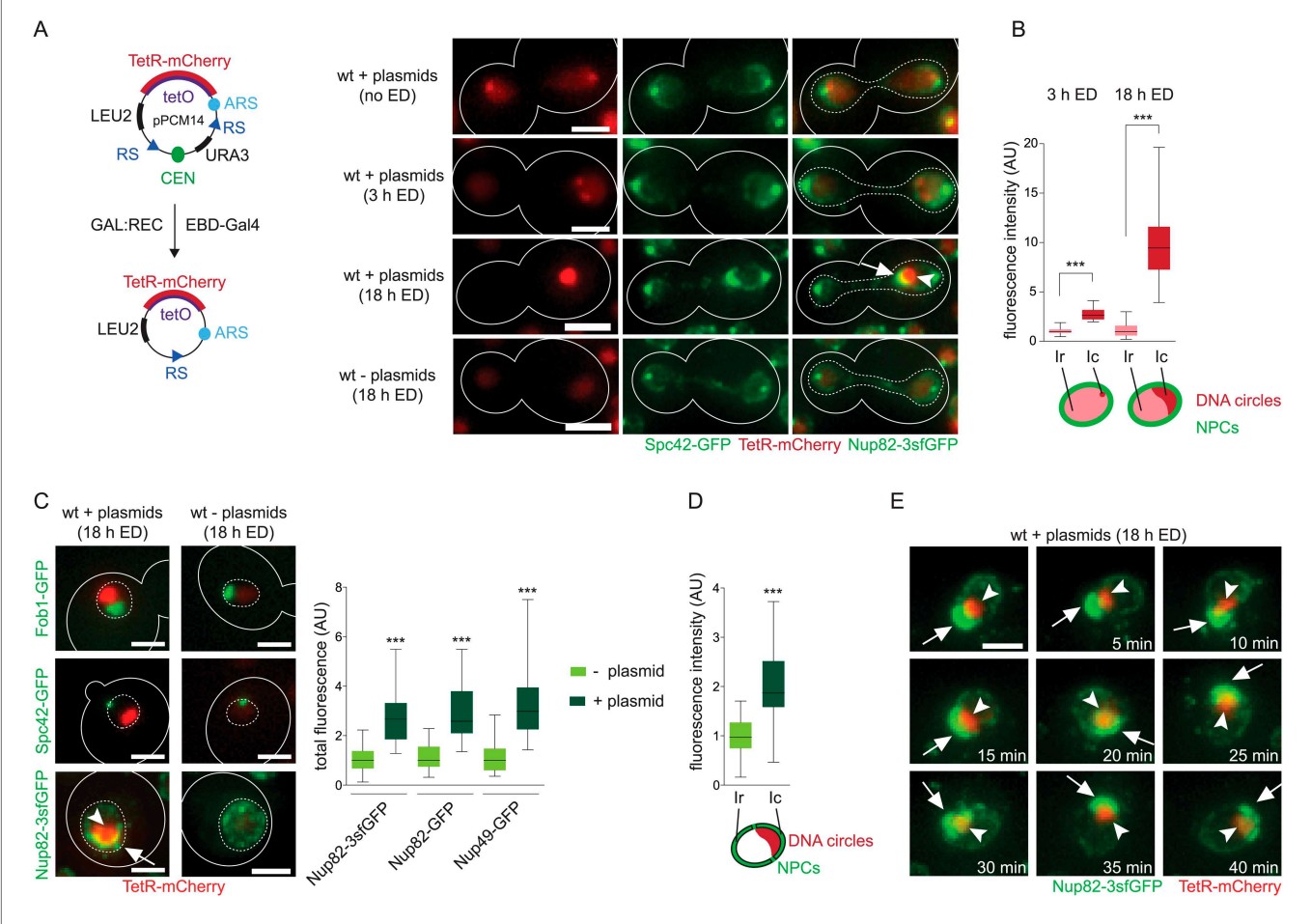

**Figure 1**. Accumulation of DNA circles induces the formation of an NPC cap. (**A**) Schematic overview of the plasmid pPCM14 and fluorescent images of pPCM14 before and 3 or 18 hr after excision of the centromere by addition of estradiol (ED) in cells expressing TetR-mCherry, Nup82-3sfGFP, and Spc42-GFP. (**B**) Quantifications of the TetR-mCherry intensity in the circle area (Ic) and the residual nuclear area (Ir) 3 or 18 hr after centromere excision (all box plots represent minimum to maximum, line represents median, N ≥ 30 cells). (**C**) Fluorescent images of the nuclear proteins Fob1, Spc42, and Nup82 (green) in cells with or without accumulated plasmids (red, 18 hr after addition of ED). Quantifications of total fluorescence of different nuclear pore markers in cells with or without accumulated plasmids, normalized by the median of cells without plasmids (N > 30 cells). (**D**) Quantifications of fluorescence intensity of Nup82 in the vicinity of the DNA circle (Ic) and the rest of the nucleus (Ir), normalized by the median Ir (N = 50 cells). (**E**) Time lapse images of a nucleus with accumulated plasmids (red) and the NPC cap (green). Images were taken every 5 min. (A-E) ***p < 0.001; images are max Z-projections; arrow depicts the NPC cap, arrow head the accumulated circles; scale bars always represent 2 μm.

increase was independent of the NPC reporter used (Nup82-3sfGFP, 2.7-folds, N = 50 cells; Nup82-GFP, 2.6-fold, N = 25 cells; Nup49-GFP, 3.0-fold, N = 25 cells, p < 0.001; *Figure 1C*) and was exclusively observed in the cells that contained DNA circles. Five generations old wild-type cells lacking plasmids did not show such changes (see Figure 9). Most strikingly, NPC distribution over the nuclear envelope was no-longer uniform; a substantial fraction of the NPCs accumulated in a single cap that covered one side of the nucleus (*Figure 1A,C*). In all cells, this cap covered the DNA circles. Furthermore, all circle-loaded cells exhibited such a cap. As a consequence, the fluorescence intensity of Nup82-3sfGFP was on an average two-fold higher in the vicinity of the circles (Ic = 2.02 ± 0.80 A.U.) compared with the rest of the nucleus (Ir = 1.0 ± 0.35 A.U., N = 50 cells, p < 0.001; *Figure 1D*). In time-lapse movies (5 min intervals for 60 min), the cap and the mCherry patch moved together (*Figure 1E*, *Video 1*), suggesting that DNA circles and the NPC cap were linked to each other. Thus, the accumulated circles failed to randomly diffuse in the nucleoplasm. Instead, they appeared to interact with some structure in the nuclear envelope, and thereby affected the morphology of the nucleus.

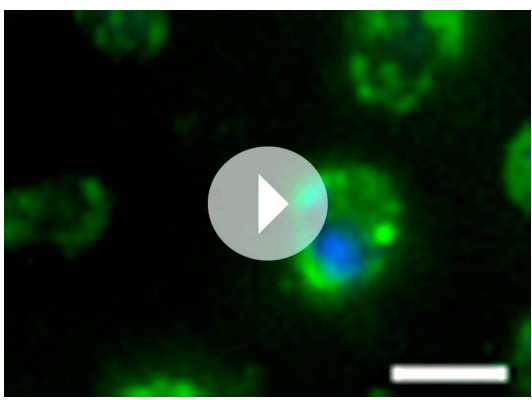

**Video 1**. The NPC cap and accumulated plasmids are simultaneously retained in the mother cell during mitosis. A cell expressing TetR-mCherry (red) and Nup82-3sfGFP (green), containing accumulated plasmids (18 hr ED), followed through nuclear division using time lapse microscopy (one Z-stack every 5 min for 1 hr, maximal projection, scale bar represents 2 μm).

## The NPC cap segregates together with the DNA circles to the mother cell during mitosis

Interestingly, these changes in NPC distribution had a clear impact on cell division, as observed in time-lapse movies (*Video 1* and *Figure 2A*). While all dividing cells formed two daughter nuclei of different sizes, the mother nucleus being generally larger, this asymmetry was enhanced in cells with accumulated circles (*Figure 2B*). Furthermore, the density of NPCs was increased in these mother nuclei (*Figure 2C*). Together, this results in an increase in the asymmetry of NPC segregation in circle-loaded cells (*Figure 2D*). Mother cells containing accumulated plasmids retained a larger fraction of their NPCs (median = 78%, N = 50), compared to cells containing no circles (63%, N = 50 cells, p < 0.001). Interestingly, in these cells both the TetR-mCherry signal and the GFP cap localized to the vicinity of the nuclear bridge linking mother and bud lobes of the dividing nucleus on its mother side (*Figures 1A and 2A* and *Video 1*). Only in very rare cases a fraction of the plasmid mass passed to the bud (2%, N > 50 cells). Qualitatively, the same effects were observed when using Nup82-3sfGFP, Nup82-GFP or Nup49-GFP as reporters. Irrespective of plasmid accumulation, tagging Nup49 slightly enhanced NPC retention (*Figure 2D*), as previously reported (*Chadrin et al., 2010*; *Makio et al., 2013*). Consistent with resulting from DNA circle accumulation, cells in which the centromere was not excised from the plasmid and untransformed cells of a similar age did not show such NPC distribution and segregation (*Figure 1* and see below). Together, these data indicate that the accumulation of non-chromosomal DNA circles affects the size of the nucleus, the organization of the nuclear envelope, and causes the mitotic retention of NPCs in yeast mother cells. These data argue against DNA circles freely diffusing in the nucleoplasm.

The diffusion barrier present in the envelope of anaphase nuclei depends on the septins and the protein Bud6 (*Shcheprova et al., 2008*). In cells lacking Bud6, the NPC cap still formed around the plasmid leading to an increase in NPC signal in the circle area (Ic = 2.29 ± 0.67 A.U.) compared to the residual area (Ir = 1.0 ± 0.33 A.U., N = 50 cells, p < 0.001; *Figure 2E*). However, in 11% of the cells, part of the plasmid mass was transmitted to the daughter cell (compared to 2% in wild-type cells, N = 50 cells). The propagated plasmid foci were generally small, but when a bigger part was transmitted, an NPC cap was visible around it (see representative image in *Figure 2E*). Thus, retention of the circles and the NPC cap partially depends on the lateral diffusion barrier separating the mother and bud parts of the nuclear envelope. Together, these data suggest that DNA circles interact with some structure at the nuclear periphery, affecting NPCs.

## The SAGA complex is required for the retention of DNA circles in yeast mother cells

Next, we wondered what caused the interaction of the circles with the nuclear periphery and if this contributed to circle retention. Therefore, we asked whether any of the protein complexes already reported to mediate chromatin interaction with the nuclear periphery facilitate the retention of DNA circles. We screened a collection of knockout strains for defects in circle partitioning, using the labeled plasmid described above. Three hours after centromere excision, individual plasmid dots were visualized and the frequency at which they segregated to the bud was determined (propagation frequency, pf, see 'Materials and methods'). In wild-type cells, non-centromeric circles rarely passed to the daughter cells (pf = 3.9 ± 1.1; *Figure 3A*), that is 2.5- to 4-fold less frequently than predicted by the passive model (pf = 10–15; (*Gehlen et al., 2011*)). Circle propagation into the bud was significantly increased in *yku70Δ* and *bud6Δ* cells, as previously reported ((*Shcheprova et al., 2008*; *Gehlen et al., 2011*); pf = 10.3 ± 1.3 and 11.8 ± 0.8, respectively, N = 3 clones, >50 cells each, p < 0.001; *Figure 3A*), validating our approach.

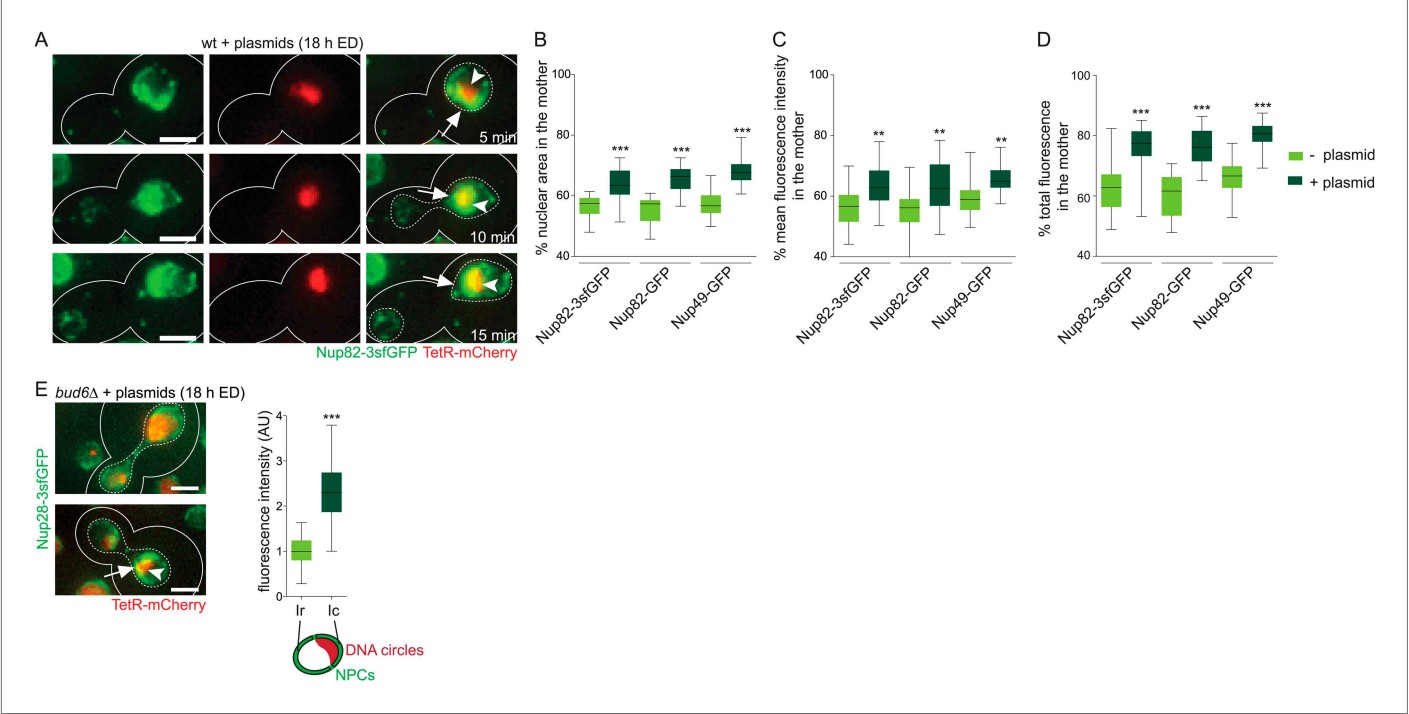

**Figure 2**. Accumulated plasmids retain the NPC cap in the mother cell during mitosis leading to an increased asymmetric segregation of NPCs. (**A**) Time lapse images of a dividing nucleus in a cell expressing Nup82-3sfGFP and containing accumulated plasmids. (**B–D**) Percentage of the nuclear area (**B**), the mean fluorescence density (**C**), and the total green fluorescence (**D**) segregated to the mother cell in telophase cells with and without plasmids using different nuclear pore markers (N > 25 cells, ***p < 0.001, **p < 0.01). (**E**) Representative pictures of *bud6Δ* cells containing plasmids and quantifications of Nup82-3sfGFP intensity in the vicinity of the DNA circle (Ic) and the rest of the nucleus (Ir) normalized by the median Ir (N = 50 cells). (**A–E**) Arrow depicts the NPC cap, arrow head the accumulated circles.

Using this assay, no effect on circle retention was observed in cells lacking any of the sirtuin proteins, Sir1-4, which are involved in chromatin silencing and telomere anchorage to the inner nuclear membrane (*Gotta et al., 1996*), in cells lacking the nucleoplasmic domain of the Sun-domain protein Mps3, involved in NPC and SPB insertion (*Jaspersen and Ghosh, 2012*) and telomere tethering to the nuclear envelope (*Bupp et al., 2007*), in cells lacking the lamin-binding-related proteins Src1 or Heh2 (*Grund et al., 2008*), and in cells lacking Slx5, a protein involved in DNA double strand break repair (*Mullen et al., 2001*) and forming perinuclear foci (*Cook et al., 2009*) (*Figure 3A*). In contrast, cells lacking Gcn5, a component of the Spt-Ada-Gcn5 acetyltransferase (SAGA) complex, showed a clear increase in plasmid propagation (pf = 11.1 ± 2.8, N = 3 clones, >100 cells each, p < 0.001), similar to the defect observed in *bud6Δ* mutant cells and fitting well with propagation frequencies predicted by the passive model. The SAGA complex regulates chromatin organization of actively transcribed genes and their recruitment to the nuclear periphery through anchorage to NPCs (*Vinciguerra and Stutz, 2004*; *Cabal et al., 2006*; *Luthra et al., 2007*). All SAGA mutants tested, including *spt3Δ* and *sgf73Δ*, displayed similar defects in DNA circle retention (pf = 9.7 ± 1.7 and 11.8 ± 1.6, respectively, N = 3 clones, >50 cells each, p < 0.001; *Figure 3A*). Furthermore, replacement of the *GCN5* gene with a point mutant allele abrogating its acetyl-ransferase activity, *gcn5-E173A* (*Wang et al., 1998*) led to a similar increase in plasmid propagation to the daughter cell as observed in *gcn5Δ* mutant cells (pf = 11.0 ± 1.5, N = 3 clones, >100 cells each, p < 0.001). Thus, SAGA and its acetyl–transferase activity, which are known to link chromatin to NPCs, promote the retention of DNA circles in the mother cell.

## SAGA mediates the retention of ERCs in yeast mother cells

In order to determine whether SAGA played a general role in circle retention, we next wondered whether it was also involved in the retention of endogenous ERCs in the mother cell. We first monitored the inheritance of an ERC marked with the *ADE2* gene (pDS163; (*Sinclair and Guarente, 1997*)) in wild-type and SAGA mutant cells, by dissecting daughters off their mothers and following the

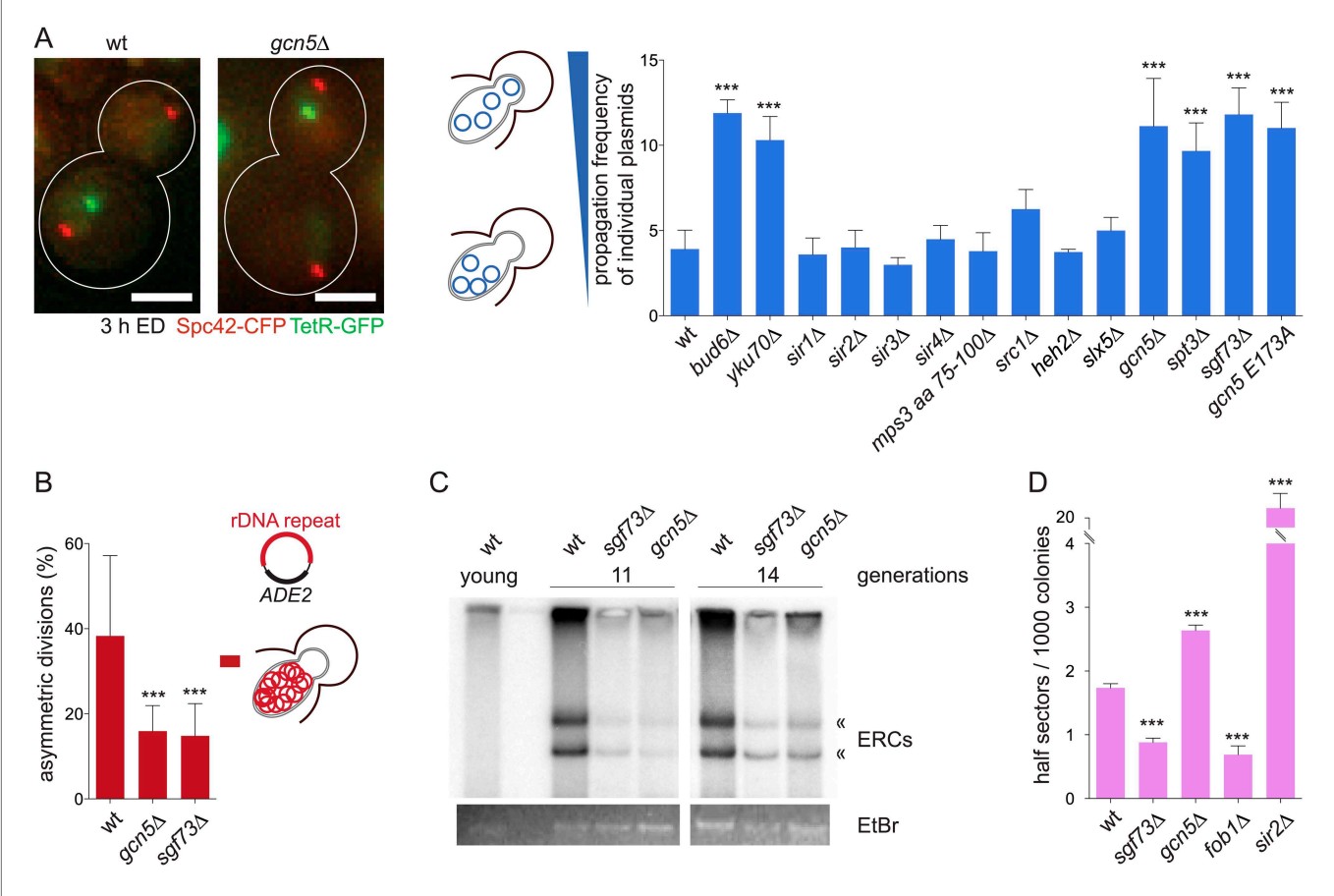

**Figure 3**. The SAGA complex mediates retention of DNA circles including ERCs. (**A**) Representative images and quantifications of plasmid propagation frequencies (pf) in wt and different mutant cells (mean ± SD N ≥ 3 clones). (**B**) Mother-bud distribution of the rDNA containing plasmid pDS163 in wt, *gcn5Δ* and *sgf73Δ* mutant cells (mean ± SD N = 10 independent experiments). (**C**) Detection of ERC levels in wt, *sgf73Δ* and *gcn5Δ* mutant cells by Southern blotting (« depict ERCs) in 11 and 14 generations old cells. (**D**) Quantifications of half-sectors representing recombination events within the rDNA repeats in wt and mutant cells (mean ± SD N = 3 clones). (**A–D**) ***p < 0.001.

inheritance of the *ADE2* marker between them. Following the segregation of the *ADE2* gene did not determine the segregation of single ERCs but solely whether all ERCs remained in the mother cell or not. In 38% (±6%) of the wild-type cells, all ERCs that had accumulated in the mother cell remained in the mother cell, as demonstrated by the fact that their daughters lacked the *ADE2* marker and formed a red colony (see material and methods, (*Sinclair and Guarente, 1997*)). This percentage was reduced 2.5-fold in the *gcn5Δ* and the *sgf73Δ* single mutant cells (16 ± 2% and 15 ± 2%, N ≥ 3 clones, p < 0.001; *Figure 3B*), indicating that high-fidelity retention of ERCs in the mother cell requires SAGA function.

To further test this conclusion, we next investigated whether the accumulation of ERCs in aged yeast mother cells depended on SAGA function. We purified wild-type, *sgf73Δ* and *gcn5Δ* mutant mother cells after 11 and 14 generations using the mother enrichment program (*Lindstrom and Gottschling, 2009*), normalized the extracted DNA using qPCR on the *ACT1* gene, and monitored ERC levels using Southern blotting. While no ERCs were detected in young wild-type cells, substantial ERC accumulation was observed in aged wild-type mothers. In contrast, SAGA mutant cells of the same age contained fewer ERCs (*Figure 3C*).

To determine whether this lower rate of ERC accumulation was due to a defect in ERC formation or ERC retention, we next estimated the impact of SAGA inactivation on recombination in the rDNA. A strain containing one rDNA repeat marked with the *ADE2* gene was used to score the rate of excision in the rDNA locus (*Jacobson and Pillus, 2009*; *McCormick et al., 2014*). Excision of the *ADE2*

marker and loss of the *ADE2* marked ERC leads to the formation of red-white sectored colonies. By scoring the frequency of half sectored colonies, we observed that the rate of ERC formation was decreased 2 folds in the *sgf73Δ*, similar but less pronounced to what *McCormick et al. (2014)* reported (*Figure 3D* and (*McCormick et al., 2014*)). However, ERC formation increased 1.5-fold in *gcn5Δ* mutant cells (*Figure 3D*). Cells lacking *SIR2* and *FOB1* served as positive and negative controls in this assay (*Gottlieb and Esposito, 1989*; *Defossez et al., 1999*; *McCormick et al., 2014*). Together, these results indicate that, at least in the *gcn5Δ* mutant mother cells, the slow rate of ERC accumulation results from the failure in their retention rather than reduced ERC formation.

Together, these data establish that SAGA promotes the retention and accumulation of both endogenous ERCs and non-chromosomal DNA circles of exogenous origins in the yeast mother cells. Whereas the function of SAGA in chromatin anchorage to NPCs was intriguing in this context, we also considered that SAGA might contribute to the retention of non-chromosomal DNA circles indirectly through its role in transcription. Therefore, we next investigated whether SAGA function was indirect, through its effect on processes already known to promote DNA circle retention in the mother cell, such as nuclear morphology, anaphase duration, and diffusion barriers.

## SAGA acts independently of the nuclear diffusion barrier and anaphase duration

Measuring anaphase duration using time lapse movies of cells expressing Nsg1-GFP revealed that deleting *YKU70* prolonged anaphase, as expected (*Gehlen et al., 2011*), whereas deletion of *BUD6*, *GCN5*, or *SGF73* did not (N ≥ 180 cells, p < 0.001; *Figure 4A*). Furthermore, nuclear bridges were not wider in *gcn5Δ* cells (N ≥ 70 cells; *Figure 4B*). Thus, we next asked whether SAGA inactivation affected the compartmentalization of the nuclear envelope during anaphase. The strength of the diffusion barrier at the bud neck of early anaphase nuclei was measured using fluorescence loss in photobleaching (FLIP), as previously described (*Luedeke et al., 2005*; *Shcheprova et al., 2008*). The barrier strength was measured in wild-type, *gcn5Δ* mutant and *bud6Δ* mutant cells, using Nup49-GFP as a reporter. Although the *bud6Δ* mutation reduced compartmentalization of the envelope, as reported previously (*Shcheprova et al., 2008*), no effect was observed in *gcn5Δ* mutant cells compared to wild-type cells (N ≥ 20 cells; *Figure 4C*). Thus, SAGA does not affect nuclear geometry, anaphase duration, and the diffusion barrier. Therefore SAGA must play a different role in circle retention.

## SAGA promotes anchorage of DNA circles to NPCs

Therefore, we next asked whether SAGA's function in tethering chromatin to nuclear pores was directly relevant for its role in circle retention. SAGA's function in chromatin anchorage to NPCs involves TREX-2, a multiprotein complex that localizes to nuclear pores, facilitates mRNA export, and physically interacts with SAGA (*Fischer et al., 2002*; *Rodriguez-Navarro et al., 2004*). Thus, we examined whether TREX-2 also contributed to circle retention. Inactivation of the core TREX-2 gene, *SAC3*, caused circles to propagate to the bud 3 times more frequently than in wild-type cells, i.e. at a frequency similar to SAGA defective cells (pf = 11.5 ± 3.0, N = 3 clones, p < 0.001; *Figure 5A*). Deletion of *SUS1*, which inactivates both SAGA and TREX-2 complexes (*Pascual-Garcia and Rodriguez-Navarro, 2009*), and deleting both the *GCN5* and *SAC3* genes led to similar effects (pf = 11.3 ± 1.4, N = 3 clones, p < 0.001 and pf = 11.1 ± 1.3, N = 3 clones p < 0.001, respectively), indicating that the effects of inactivating SAGA and TREX-2 were not additive. We conclude that the SAGA complex and the NPC component TREX-2 act together in a pathway mediating the targeted retention of non-chromosomal circles in the mother cell.

Thus, we next wondered whether TREX-2 and SAGA promoted the linkage of DNA circles to NPCs (see model *Figure 5A*). In order to investigate this possibility, we first tested whether individual non-centromeric plasmids co-localize with nuclear pores and whether this required SAGA function. Consistent with this idea, the vast majority of non-centromeric circles localized to the nuclear rim in wild-type cells: only 15% (±2.8%) of plasmid foci were at a resolvable distance from the nuclear envelope (*Figure 5B*). In contrast, this fraction was substantially increased in cells lacking SAGA (*gcn5Δ* mutant cells) or both SAGA and TREX-2 function (*sus1Δ* mutant cells; 31.9% ± 1.3% and 24.1% ± 2.6%, respectively, N = 3 clones, p < 0.001). To determine whether the plasmids localizing to the periphery co-localized with the nuclear pores, Nup82-3sfGFP intensity traces (N > 50 cells) were measured along the nuclear envelope in equatorial focal sections of the nuclei containing a single labeled acentric DNA circle at the rim. High LED-based illumination and fast image acquisition minimized blur due to

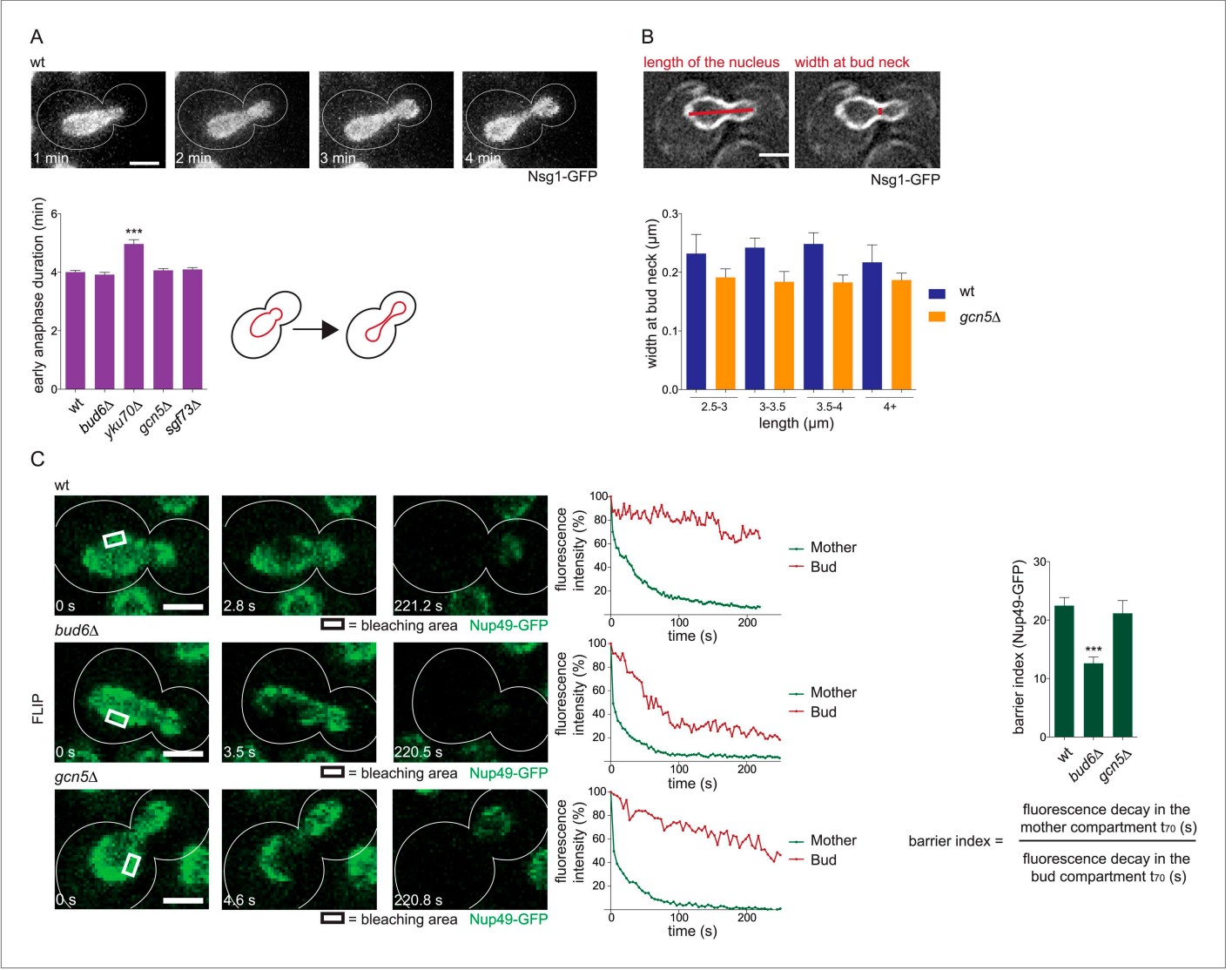

**Figure 4**. Previously described mechanisms for plasmid retention are not affected in SAGA deficient cells. (**A**) Duration of early anaphase in wt cells and cells lacking Bud6, Yku70, Gcn5, or Sgf73 (mean ± SD N ≥ 180 cells). (**B**) Measurements of the width of early anaphase nuclei at the bud neck, categorized by different length of the nucleus in wt and *gcn5Δ* mutant cells (mean ± SEM, N ≥ 70 cells). (**C**) Photobleaching analysis of wt, *bud6Δ*, and *gcn5Δ* cells expressing Nup49-GFP. Graph shows the Barrier Index (ratio of the time to decay to 70% of initial fluorescence in the mother compartment to the bud compartment of early anaphase nuclei, mean ± SEM, N ≥ 20 cells). (**A**–**C**) ***p < 0.001. Images are max Z-projections (**A**) or represent one focal plane (**B** and **C**).

NPC movement. Aligning all traces with the position of the DNA circle showed that the average Nup82-3sfGFP intensity increased in the vicinity of the circle compared to elsewhere (+33.2% ± 0.09%, *Figure 5C*). No such increase in nuclear pore signal was observed near DNA circles in *gcn5Δ* and *sus1Δ* mutant cells (+4.6% ± 0.05% and −2.0% ± 0.07%, p < 0.01). Thus, we conclude that SAGA indeed promotes the interaction of non-chromosomal DNA circles to nuclear pores.

## Artificial tethering of circles to pores bypasses the need for SAGA

SAGA is involved in many different processes, including transcription, thus we wondered whether detachment from the pores was the main reason for the increased circle propagation observed in SAGA deficient cells. To test this, we artificially tethered circles to the NPC components Nup49 or Nup170, as previously reported (*Khmelinskii et al., 2011*), and asked whether this was sufficient to bypass the requirement for SAGA function in circle retention. Neither fusion protein had any effect on

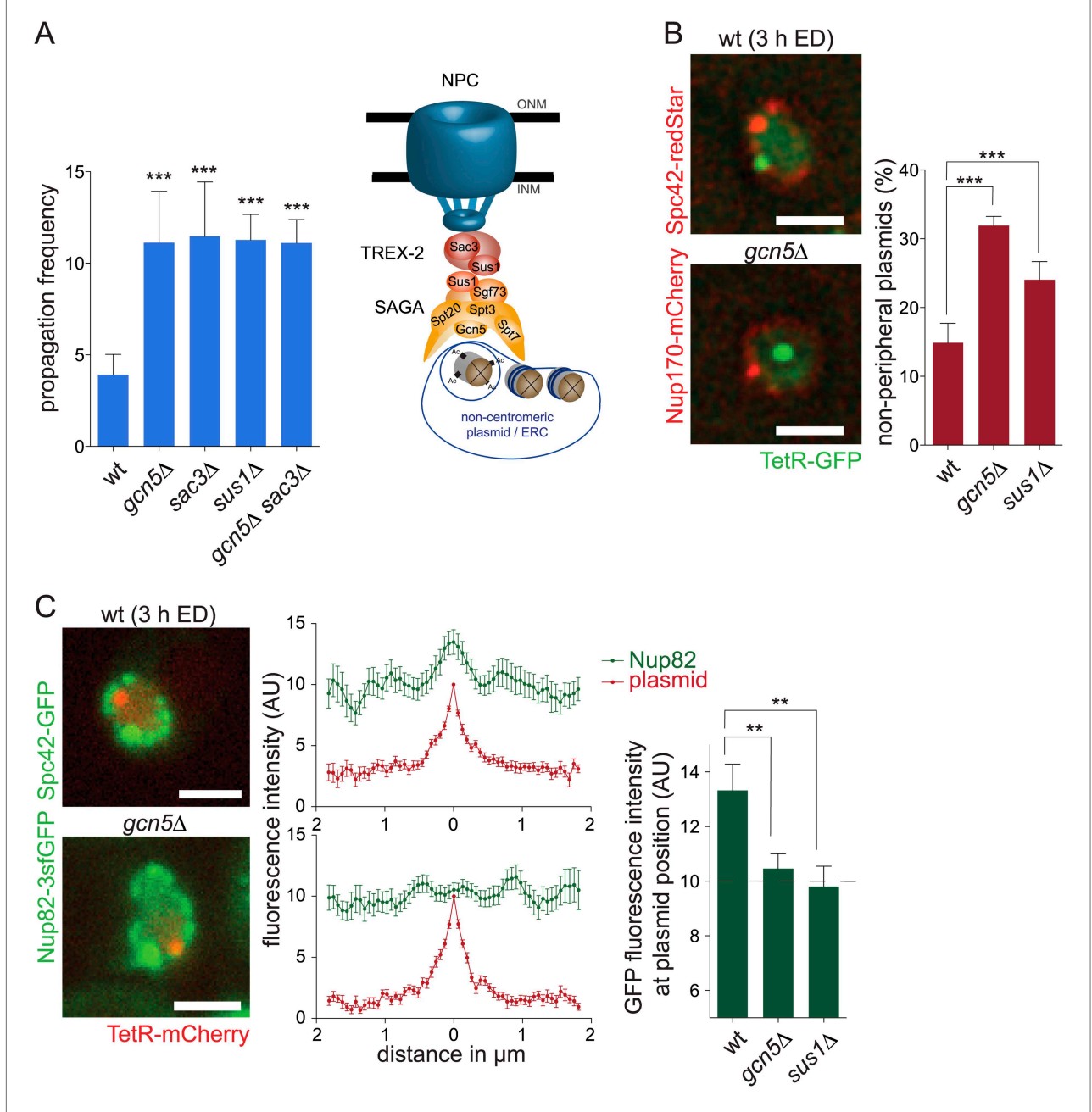

**Figure 5**. SAGA attaches DNA circles to NPCs via TREX-2. (**A**) pf in wt and different mutant cells (*gcn5Δ* from **Figure 3A** for comparison, mean ± SD, N ≥ 3 clones). Model of how non-centromeric plasmids including ERCs might be attached to NPCs via the SAGA and TREX-2 complexes. (**B**) Percentage of plasmids at a resolvable distance to the nuclear periphery in wt, *gcn5Δ* and *sus1Δ* mutant cells (mean ± SD, N = 3 clones). (**C**) Normalized fluorescence intensity of Nup82-3sfGFP in green (NPCs) and TetR-mCherry in red (plasmid) aligned by the peak of TetR intensity. Quantifications of average Nup82-3sfGFP intensity 130 nm around the plasmid (mean ± SEM, N = 50 cells). (A-C) ***p < 0.001, **p < 0.01. Images represent one focal plane.

circle retention when replacing the corresponding endogenous nucleoporin in otherwise wild-type cells (pf = 5.5 ± 0.7 in *NUP170-TetR* and 4.6 ± 1.6 in *NUP49-TetR* cells; **Figure 6A**), as reported. In contrast, when expressed in *gcn5Δ*, *sgf73Δ*, or *sus1Δ* mutant cells, these fusion proteins restored the retention of the reporter circle in the mother cell fully or largely so. In support of the TetR-fusion proteins mediating circle attachment to NPCs, the labeled circles localized close to the envelope in these cells (**Figure 6B**). Importantly, tethering of the circles to the nuclear pores did not bypass the requirement for the diffusion barrier in plasmid retention, since *bud6Δ* mutant cells expressing Nup170 or

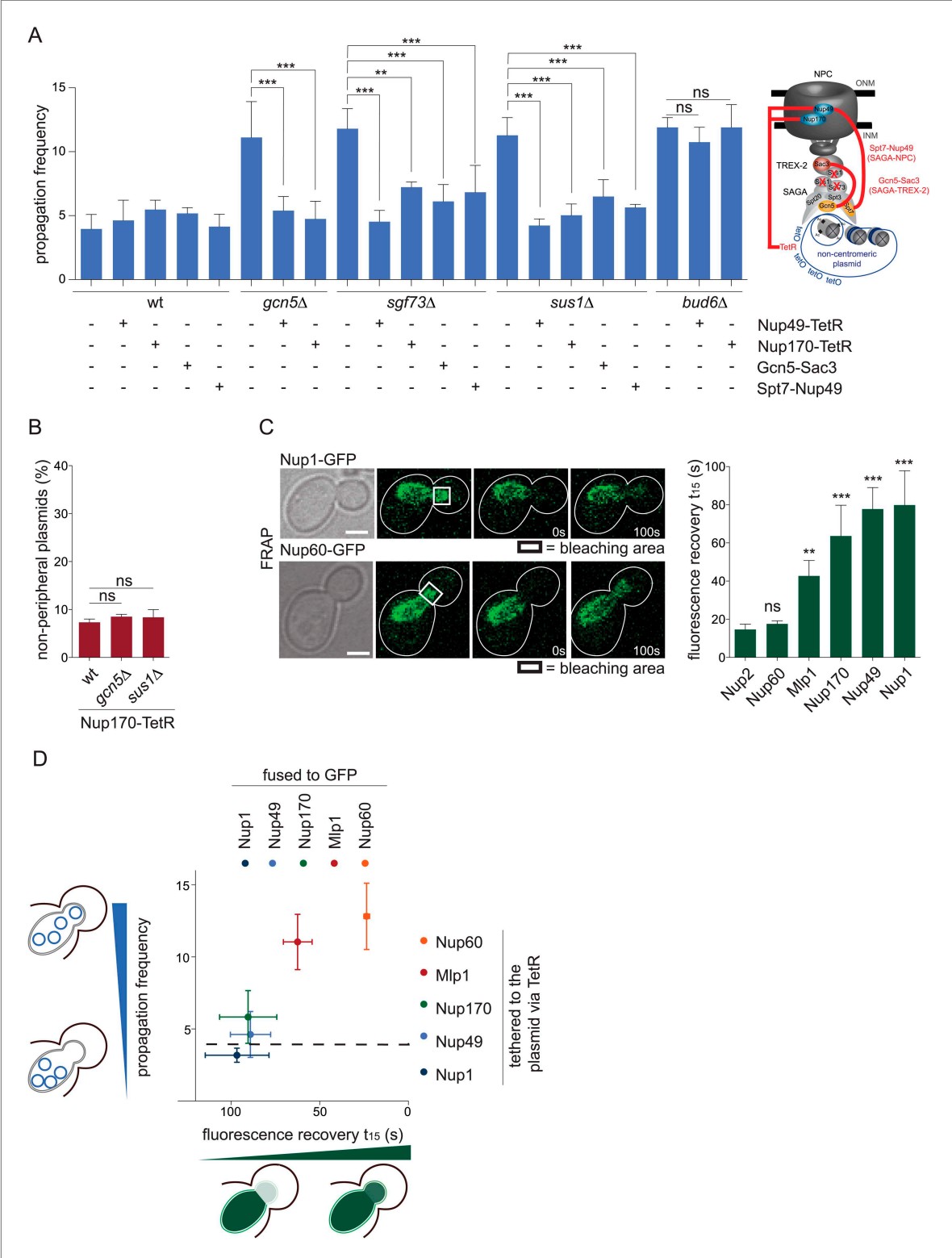

**Figure 6**. SAGA-dependent attachment of circles to stable NPC components ensures their asymmetric segregation. (**A**) pf in wt and different mutant cells expressing Nup170 or Nup49 fused to TetR or the fusion proteins Gcn5-Sac3 or Spt7-Nup49 (mean ± SD, N ≥ 3 clones). Scheme of the fusion proteins. (**B**) Percentage of plasmids at a resolvable distance to the nuclear periphery in wt, *gcn5Δ* and *sus1Δ* mutant cells expressing Nup170 fused to TetR (mean ± SD, N = 3 clones) (**C**) Fluorescence recovery after bleaching the bud part of early anaphase nuclei measured in cells expressing several

*Figure 6. Continued on next page*

*Figure 6. Continued*

NPC components tagged with GFP. Quantification of the time to recover 15% of fluorescence intensity ($t_{15}$, mean ± SEM, all compared with Nup2 for statistics). (**D**) Correlation between the fluorescence recovery in the bleached bud compartment ($t_{15}$) of the depicted proteins fused to GFP and the pf in cells expressing the same proteins fused to TetR. Dashed line represents pf in wt cells. (**A–C**) ***$p < 0.001$, **$p < 0.01$.

Nup49 fused to TetR displayed plasmid propagation frequencies similar to *bud6Δ* cells (pf = 11.9 ± 1.8 and 10.8 ± 1.2, respectively; *Figure 6A*). Thus, non-chromosomal DNA circles artificially tethered to NPCs no longer need SAGA for being efficiently retained in the mother cell by the diffusion barrier.

The study by *Khmelinskii et al. (2011)* indicated that anchoring of a non-centromeric plasmid to 11 different NPC components had no effect on plasmid retention, but that targeting of the two basket proteins Mlp1 and Nup2 to the reporter plasmid was sufficient to impair its retention in the mother cell (*Khmelinskii et al., 2011*). We reasoned that the first set of observations is expected if circles are already tethered to the same NPCs by SAGA. To better understand the effect of the two other fusion proteins, we first investigated whether they localized stably at NPCs, using fluorescence recovery after photobleaching (FRAP; *Figure 6C*). Indeed, anchorage would require a stable interaction, whereas at least Nup2 has been reported to shuttle between NPCs and the nucleoplasm (*Dilworth et al., 2001*). Accordingly, we found that both Nup2 and Mlp1, as well as Nup60, diffuse more rapidly than the core nucleoporins Nup1, Nup49, and Nup170, at least during anaphase (*Figure 6C*), indicating that their residence at NPCs is transitory. Remarkably, the stable association of nucleoporins with NPCs was highly correlated with their ability to mediate circle retention (*Figure 6D*). Together, our data indicate that artificially attaching non-centromeric plasmids to NPC components bypasses the need for SAGA function, provided that the used nucleoporin is stable at NPCs. Thus, SAGA functions in retention through promoting stable anchorage of the circles to NPCs.

To investigate how direct the effect of SAGA on circle anchorage to NPCs might be, we next asked whether a physical link between SAGA and NPCs was required for SAGA's function in mediating the anchorage of circles to NPCs and whether SAGA interacted with the DNA circles themselves. To address the first question, we generated fusion proteins to covalently link the SAGA complex to the TREX-2 complex (Gcn5-Sac3) or directly to NPCs (Spt7-Nup49) and asked whether this was sufficient to restore the retention of DNA circle in cells lacking Sgf73 or Sus1. Expression of either of these two fusion proteins had no effect on plasmid retention in otherwise wild-type cells (pf = 5.2 ± 0.4 and 4.1 ± 1.0, respectively; *Figure 6A*). However, it largely bypassed the need for Sgf73 (pf = 6.1 ± 1.3 with Gcn5-Sac3 and 6.8 ± 2.0 with Spt7-Nup49, compared to 11.8 ± 1.6 in *sgf73Δ* cells) and *Sus*1 (pf = 6.5 ± 1.3 with Gcn5-Sac3 and 5.6 ± 0.2 with Spt7-Nup49, compared to 11.3 ± 1.4 in *sus1Δ* cells, N = 3 clones, p < 0.001). Thus, the main function of Sgf73 and *Sus*1 in plasmid retention is to physically link SAGA to nuclear pores. These data establish that SAGA needs to be at NPCs in order to function in circle retention.

## SAGA preferentially binds non-centromeric DNA molecules

To address whether SAGA interacts with non-centromeric DNA circles, we then analyzed the localization of Gcn5-GFP and Sgf73-GFP fusion proteins in cells accumulating non-centromeric DNA circles labeled with TetR-mCherry. Both SAGA components were strongly enriched in the plasmid area, indicating that SAGA interacts more readily with non-chromosomal DNA than with the rest of the genome (*Figure 7A*). Accordingly, chromatin immunoprecipitation (ChIP) experiments (N = 5 independent experiments) reproducibly showed a clear enrichment of the SAGA proteins Gcn5 and Spt20 on a non-centromeric reporter plasmid (pYB1670/CEN-), as judged by probing for the autonomous replication sequence (ARS) and two non-coding sequences of bacterial origin. Strikingly, this enrichment was nearly lost (p < 0.05) when the same plasmid contained a centromere (CEN+; *Figure 7B*). Thus, SAGA is recruited to higher levels on non-centromeric than on centromeric DNA circles. Together, these data indicate that SAGA more stably associates with non-chromosomal than with chromosomal DNA.

To further test the role of SAGA recruitment on the circles in circle retention, we tested the effect of increasing SAGA levels on the plasmid. We increased Gcn5 levels on our TetO circle through expression of a Gcn5-TetR fusion protein and asked whether this affected its retention. Strikingly, this construct reduced the propagation of the circle to the bud 3-fold compared to wild-type (pf = 1.1 ± 0.3; N = 3 clones, p < 0.001; *Figure 7C*). In contrast, cells expressing the deacetylase Sir2 fused to TetR

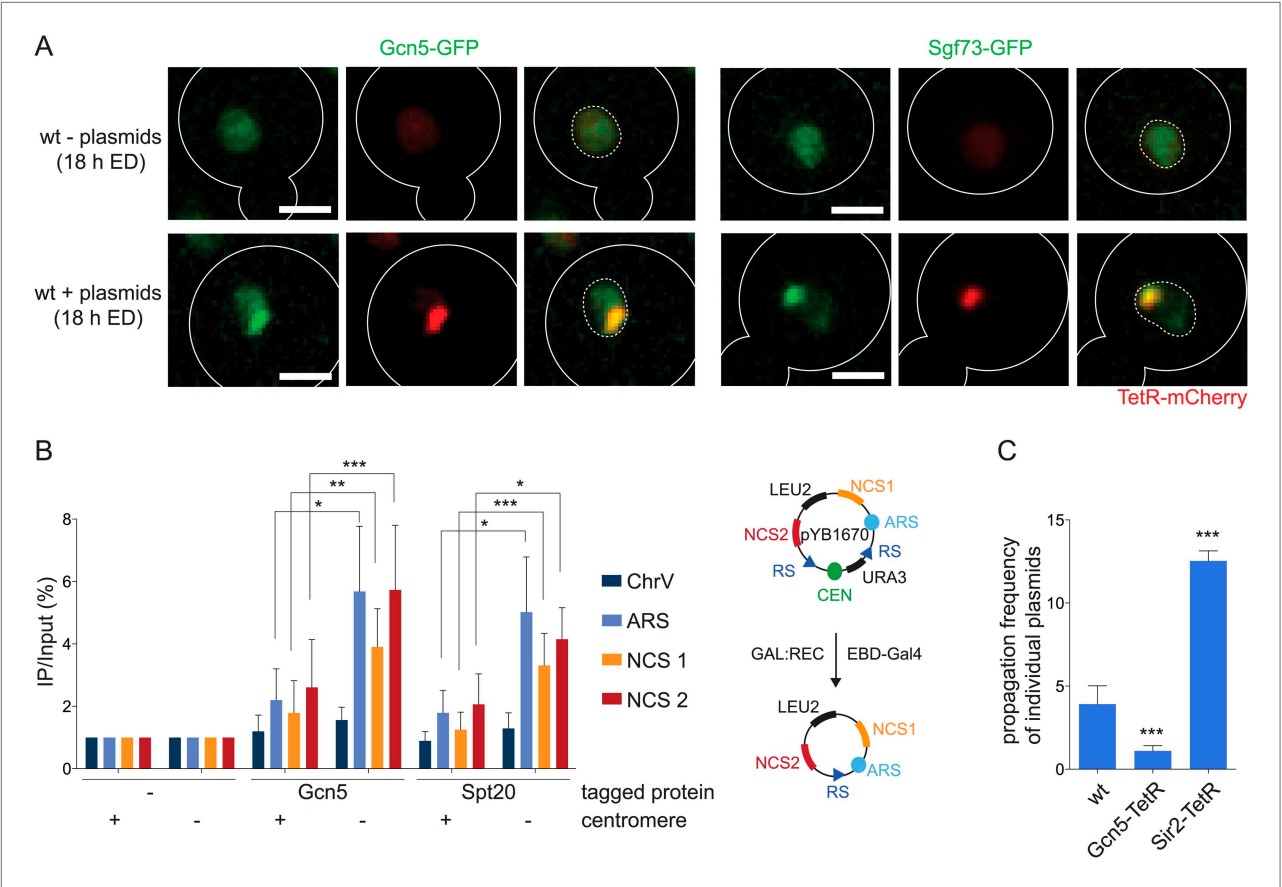

**Figure 7**. SAGA preferentially interacts with non-centromeric DNA circles (**A**). Fluorescent images (deconvolved max Z-projections) of Gcn5-GFP and Sgf73-GFP (green) in cells with and without accumulated plasmids (red). (**B**) ChIP-qPCR analysis to test the binding of the SAGA components Gcn5 and Spt20 to three sequences on pYB1670 (ARS, non-coding sequences (NCS) 1 and 2; see scheme of plasmid) and a control sequence on chromosome V (mean ± SD, N = 5 independent experiments, ***p < 0.001, **p < 0.01, *p < 0.05). (**C**) pf in cells expressing Gcn5 or Sir2 fused to TetR (mean ± SD, N = 3 clones, ***p < 0.001).

segregated the circle to the bud as frequently as SAGA defective cells. We concluded that artificial loading of Sir2 to the plasmid either interfered with normal SAGA recruitment to the plasmid or that Sir2 constantly counteracts SAGA's function on the plasmid. Thus, we concluded that binding of SAGA to the circle is required for efficient retention of the circle in the mother cell.

## Acentric DNA circles and nuclear pores influence each other's dynamics in a SAGA-dependent manner

So far, our data have established that SAGA promotes the attachment of non-chromosomal DNA circles to NPCs and that this attachment is essential for the efficient retention of the circles in the mother cell. This observation is at odds with data by several labs indicating that NPCs are not nearly as well retained in the mother cell than DNA circles (*Khmelinskii et al., 2010*; *Colombi et al., 2013*; *Menendez-Benito et al., 2013*). Possibly accounting for these conflicting data, we observed above (*Figure 2A*) that the mother cells that were loaded with circles retained more pores. Thus, we reasoned that DNA circles might promote the retention of the NPCs to which they were attached. In favor of this idea, and in contrast to what we observed in wild-type cells (*Figure 2A*), the accumulation of DNA circles had no effect on pore segregation in *gcn5Δ* and *sgf73Δ* single mutant cells (*Figure 8A*). Thus, SAGA-dependent attachment of the circles to pores promotes the retention not only of the circles but also of those NPCs.

To shed light on why circle-bound NPCs are better retained in the mother cell, we next asked whether these NPCs showed different dynamic behavior as unbound NPCs. FLIP experiments revealed

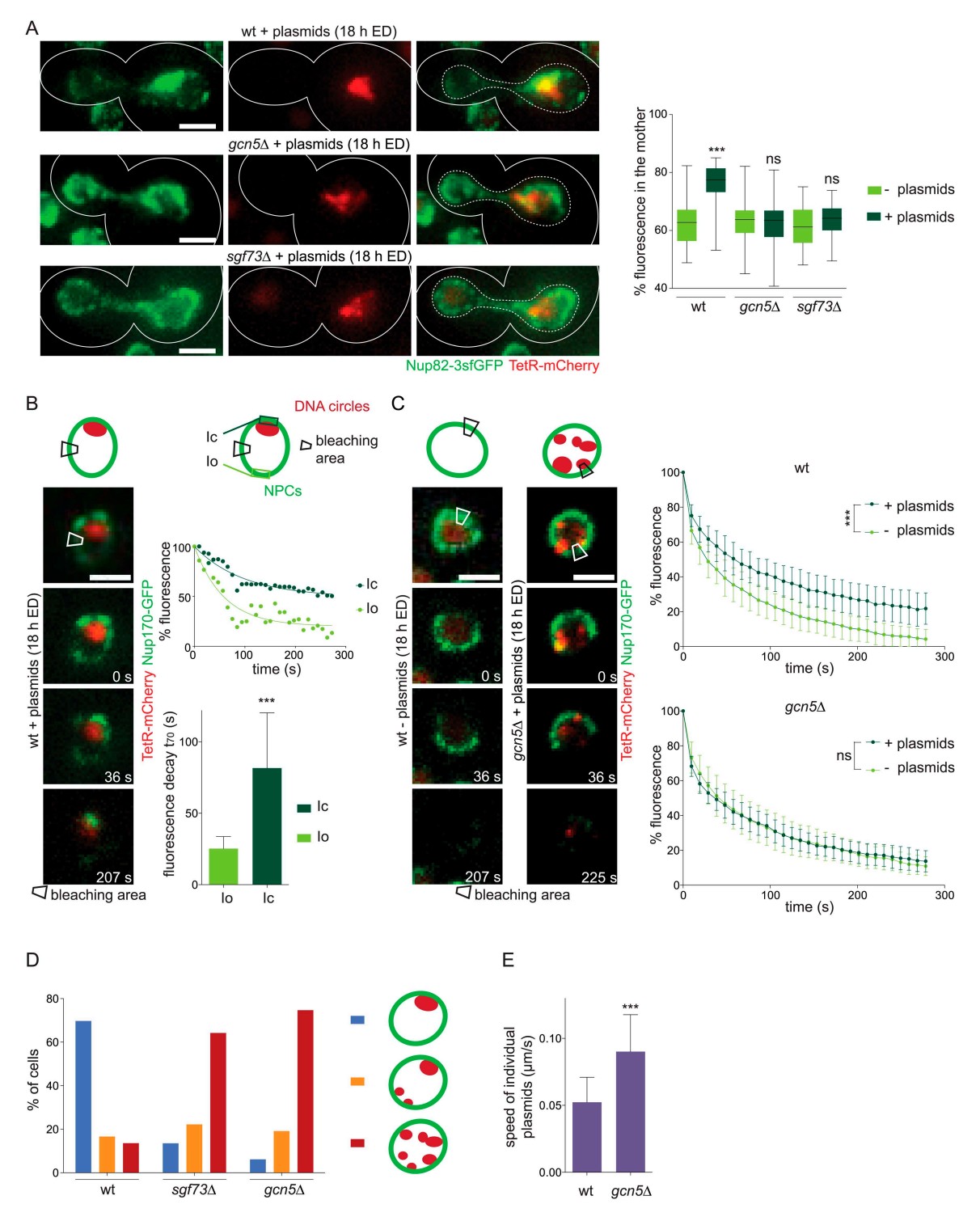

**Figure 8**. DNA circles and NPCs reciprocally reduce their dynamics in a SAGA-dependent manner. (**A**) Fluorescent images of Nup82-3sfGFP (green) in cells with accumulated plasmids (red). Quantification of the percentage of NPCs segregated to the mother cell in wt, *gcn5Δ* and *sgf73Δ* cells with or without plasmids (box plots: min to max, median (line), N = 50 cells). Images are max Z-projections. (**B**) Time lapse images of a photobleaching experiment in wt cells expressing Nup170-GFP (green) and containing plasmids (red). Rectangle depicts the bleaching area. Plotted Nup170-GFP intensity in the circle area (Ic) and an equidistant area opposite of the circles (Io) over time, set to 100% prior to bleaching. Images represent one focal plane. Quantification of the time 30% of fluorescence decays ($t_{70}$; mean ± SD, N = 20 cells). (**C**) Average total fluorescence of the whole nucleus plotted over

*Figure 8. Continued on next page*

*Figure 8. Continued*

time in wt and *gcn5Δ* mutant cells with (dark green) and without plasmids (light green), set to 100% prior to bleaching (mean ± SD, N ≥ 20 cells, ***p < 0.001). (**B** and **C**) Images represent one focal plane. (**D**) Quantification of plasmid clustering in wt and SAGA deficient cells. Cells were divided in 3 categories: plasmids localized always, partially, or never to one focus throughout a 1 hr time lapse movie (mean, N ≥ 160 cells). (**E**) Speed of individual plasmids in wt and *gcn5Δ* cells expressing TetR-GFP measured from time lapse movies with 3 s intervals (mean ± SD, N ≥ 50 plasmids). (**A–E**) ***p < 0.001.

that SAGA-mediated attachment of circles to NPCs strongly affected their diffusion speed. When fluorescence intensity was measured both in the vicinity of the circles (Ic) and at opposite side of the nucleus (Io) in wild-type cells loaded with circles, with both measurement regions equidistant from the bleached spot, the signal decayed slower and to a lesser extent in regions associated with the circles ($t_{70}$ = 82 ± 39 s) than in those on the opposite side ($t_{70}$ = 25 ± 8 s, N = 20 cells, p < 0.001; *Figure 8B*).

We could not perform the same experiment in SAGA deficient cells because circles did not cluster to one focus and the cells formed no NPC cap. NPC distribution remained uniform over the nuclear surface, and non-centromeric circles were scattered throughout the nucleoplasm (*Figure 8D*). We concluded that SAGA mediates both circle clustering and the formation of the NPC cap. Accordingly, in FLIP experiments the fluorescence decayed homogenously over the entire nuclear envelope, whether they contained circles or not ($t_{50}$ = 46 ± 9.1 s for *gcn5Δ* and 41 ± 6.7 s for *sgf73Δ* mutant cells; *Figure 8C*), as in wild-type cells devoid of circles ($t_{50}$ = 35 ± 5.9 s). Thus, SAGA-dependent attachment of non-centromeric DNA circles substantially, and specifically, reduces the mobility of the corresponding NPCs. These data establish two points. First, in circle-free exponentially growing cells only very few NPCs, if any, are stably attached to and slowed down by chromatin. Second, NPCs that are attached to DNA circles show very slow dynamics, possibly explaining their enhanced retention in the mother cell during mitosis.

Consistent with the attachment of circles to NPCs affecting each other's dynamics, SAGA also slowed down the non-centromeric DNA circles. In time-lapse movies (one stack every 3 s, 20 optical slices per stack, for 3 min), the average speed of the DNA circles was significantly higher in cells lacking Gcn5 (0.09 ± 0.027 μm/s, N ≥ 50 plasmids, p < 0.001) than in wild-type cells (0.05 ± 0.019 μm/s; *Figure 8E*). Thus, our data establish that non-chromosomal DNA circles and NPCs reciprocally constrain the diffusion speed of each other, in a SAGA-dependent manner.

## Asymmetric segregation of NPCs increases with age leading to their accumulation in old cells

Together, our data suggest that SAGA targets non-chromosomal DNA circles and mediates their retention in the mother cell by linking them to NPCs. However, since these results were all obtained with reporter plasmids, we next asked whether SAGA function was also relevant for endogenous circles. We reasoned that, if ERCs rely on an interaction with NPCs for their retention in the mother cell, then the NPCs attached to these ERCs should also be retained and accumulate with age. To test this possibility, we monitored nuclear pores in ageing cells using Nup49-, Nup170-, or Nup82-GFP fusion proteins as a reporter. To allow imaging them throughout their lifespan, the cells were grown in a microfluidic device (*Lee et al., 2012*) that selectively retains the mother cells, due to their larger size, while their daughter cells are removed by the medium flow. Transmission pictures were taken for 64 hr at 20 min intervals, providing information about the age of all cells present (*Video 2*). In addition, 2h fluorescence movies at 15 min intervals were recorded after 48 hr and 64 hr. At both time points, the original cells and some coincidentally trapped cells born in the chip were present, allowing simultaneous monitoring of pores in cells of different ages (ranging from 0 to 38 generations; *Figure 9A*). Strikingly, a large majority of the aged cells showed a non-uniform distribution of NPCs on the nuclear surface. A substantial fraction of very old cells formed an even more intense cap (*Figure 9B*). These cells divided maximally three more times before dying. Thus, cap formation was observed upon ageing in wild-type cells.

To characterize how NPC inheritance evolved as the cells aged, the total fluorescence of the Nup49-, Nup170-, and Nup82-GFP reporters was measured in image stacks through the entire volume of mother and daughter nuclei of telophase cells and plotted as a function of the age of the mother cell. For all three NPC markers, fluorescence intensity increased significantly in the mother nuclei with

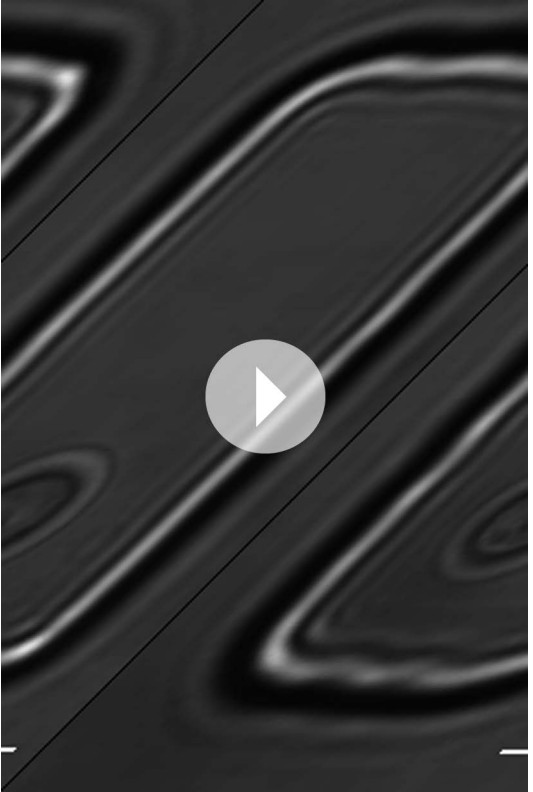

**Video 2**. A microfluidic device to follow individual cells throughout their entire life. A trapped cell is visualized for 67 hr taking a transmission picture every 20 min. The trapped cell reaches age 29 before it dies. Coincidently, other cells are trapped under the same micropad, whereas most daughter cells are washed away by a constant flow of media.

age (*Figure 9C*). This effect was strongest for Nup49-GFP, as above (*Figure 2*). The increasing amount of NPCs in the mother cell did not happen at the cost of the daughter cells. On the contrary, occasionally an increased amount of nuclear pores was also observed in daughter cells of old mother cells (>15 generations; *Figure 9C*). Thus, like in young cells loaded with artificial circles, the number of NPCs increases in ageing mother cells whereas the number of pores transmitted to the daughter cells remains fairly constant.

Accordingly, the fraction of nuclear pores that segregated towards the mother cell during mitosis increased as the cells aged, indicating that these cells partitioned NPCs in an increasingly asymmetric manner (*Figure 9D*). Young mother cells expressing Nup170-GFP or Nup82-GFP retained 56 ± 1.9% and 57 ± 2.5% of the signal in the mother cell. The retention of pores in the mother cell increased after 25 generations to 73 ± 5.0% (Nup82-GFP) and 78 ± 3.8% after 35 generations (Nup170-GFP; *Figure 9D*). Cells expressing Nup49-GFP showed the same trend, although asymmetry was higher in both young (63% ± 3.3%) and older cells (81% ± 16.1% after 20 generations; *Figure 9D*). Analysis of nuclei in G1 cells of increasing age confirmed that old mothers contained more NPCs than young cells (on average 4.7 ± 1.2-fold more than young mothers in cells expressing Nup49-GFP after 25 generations, 4.0 ± 1.1-fold in Nup170-GFP expressing cells after 30 generations, and 4.1 ± 1.4-fold in Nup82-GFP expressing cells after 35 generations; *Figure 9E*). These nuclei were both larger (radius = 1.5 fold larger; *Figure 9F*) and more densely populated with pores (1.9-fold increase in density; *Figure 9G*) compared to the nuclei of young cells. Remarkably, the increase with age in NPC asymmetry, NPC retention, and NPC number was much slower in cells lacking a diffusion barrier (*bud6Δ* mutant cells; *Figure 9C–E*). Thus, consistent with the possibility that ERC accumulation affects NPC segregation, mother cells retain an increased number of NPCs as they age, in a diffusion barrier-dependent manner.

## ERC accumulation and SAGA promote the age-dependent accumulation of NPCs

In order to test whether NPC accumulation in old cells indeed depended on ERCs, we next characterized the NPC content of ageing *sir2Δ* mutant cells, which form ERCs earlier, and *fob1Δ* mutant cells, which form ERCs much later (*Defossez et al., 1999*; *Kaeberlein et al., 1999*). Remarkably, the proportion of NPCs retained in the mother cell at mitosis and the total amount of pores in G1 cells increased much slower with age in *fob1Δ* mutant cells compared to wild-type, irrespective of the reporter used (Nup49-GFP or Nup170-GFP), and faster in the *sir2Δ* mutant mother cells (*Figure 10A–C*). Thus, the presence of ERCs largely drives NPC retention in ageing cells.

Consistent with the SAGA-dependent effect of non-centromeric DNA circles on pore segregation in young cells, the *sgf73Δ* mutant cells accumulated NPCs slowly as they aged (*Figure 10A–C*). Furthermore, artificially linking SAGA to NPCs using the Gcn5-Sac3 and Spt7-Nup49 fusion proteins restored NPC accumulation in ageing *sgf73Δ* mutant cells (*Figure 10D*). Based on these and our findings above, we conclude that SAGA mediates the interaction of ERCs and NPCs, leading to the joined retention and accumulation of both ERCs and NPCs in ageing yeast mother cells.

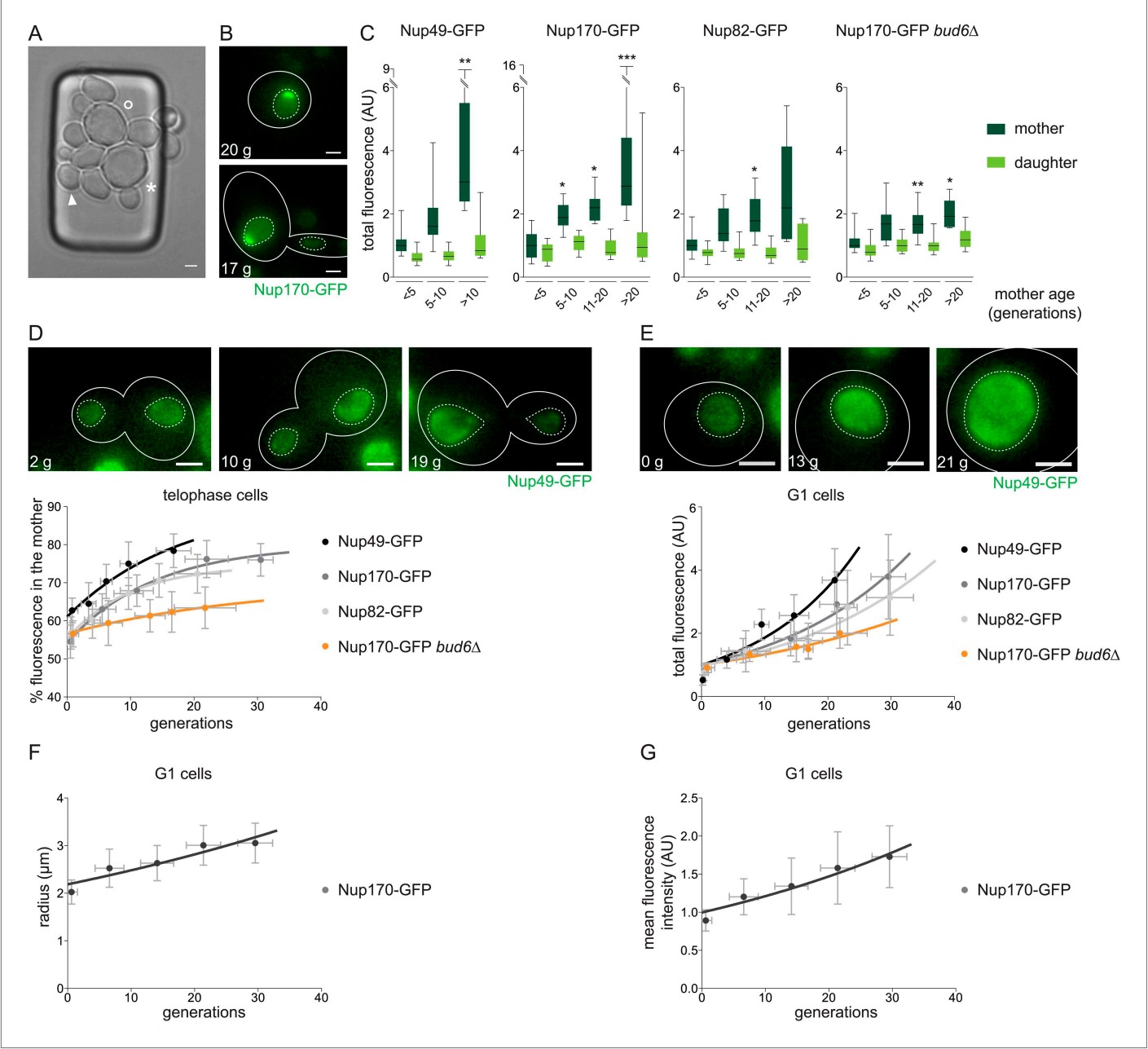

**Figure 9**. NPCs segregate increasingly asymmetric with age leading to their accumulation in old cells in a barrier dependent manner. (**A**) Transmission image of cells trapped in the microfluidic device after 64 hr. Cells of different age are trapped (➤, °, and * depicts 0-, 27-, and 31-generation old cells, respectively). (**B**) Cells expressing Nup170-GFP showing a pronounced NPC cap. (**C**) Total fluorescence of the depicted NPC marker in the mother (dark green) and daughter cell (light green) grouped by age categories of the mother cells (N ≥ 50 cells, ***p < 0.001, **p < 0.01, *p < 0.05). (**D**) Percentage of NPC fluorescence segregated to the mother cell plotted against their age in wt cells expressing different NPC markers and *bud6Δ* cells. Lines show fitted curves; dots represent the average of 10 data points grouped by age (mean ± SD, N ≥ 50 cells). (**E–G**) Quantifications of total fluorescence (**E**), nuclear radii (**F**), and mean fluorescence intensity (**G**) in G1 cells of increasing age (mean ± SD, N ≥ 50 cells) quantified as in (**D**). (**A–E**) Numbers in white depict the age of the corresponding cell (g = generations). Images are sum Z-projections.

## The SAGA complex promotes ageing in yeast

The retention of DNA circles contributed to the replicative ageing in yeast and SAGA deficient cells accumulated less ERCs and NPCs. Therefore, we next wondered whether these cells lived longer. Using microdissection, the lifespan of cells lacking Gcn5 and Spt20, a structural component of the SAGA complex, as well as Sgf73, linking SAGA to TREX-2, was determined. Wild-type cells showed

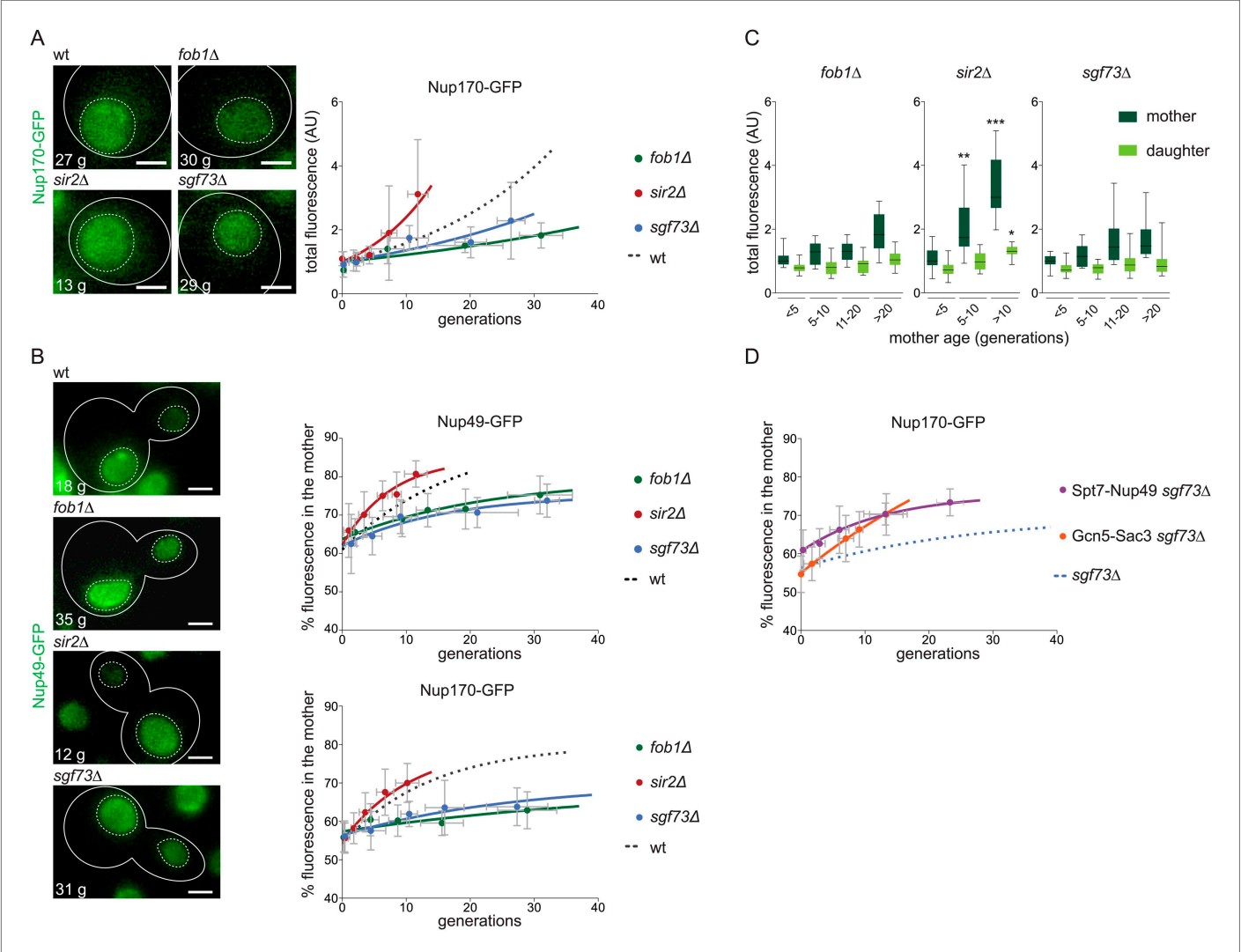

**Figure 10**. The accumulation of NPCs in old mother cells depends on ERCs and SAGA. (**A**) Total Nup170-GFP intensity in G1 cells plotted against their age in wt and depicted mutant cells. (**B**) Percentage of fluorescence segregated to the mother cell in wt and depicted mutant cells expressing Nup170-GFP or Nup49-GFP. (**A** and **B**) Images are sum Z-projections. (**C**) Total fluorescence of Nup170-GFP in mother (dark green) and daughter cells (light green) grouped by age categories of the mother cells (N ≥ 50 cells, ***p < 0.001, **p < 0.01, *p < 0.05). (**D**) Percentage of total Nup170-GFP fluorescence segregated to the mother cell in cells lacking Sgf73 and expressing either Spt7-Nup49 or Gcn5-Sac3. (**A**, **B**, **D**) Lines show fitted curves; dashed lines for comparison; dots represent the average of 10 data points grouped by age (mean ± SD, N ≥ 50 cells).

a median lifespan of 27 generations, whereas the *sgf73Δ* mutant cells lived 60% longer, (44 generations, N = 150 cells, p < 0.001; *Figure 11A* (*Schleit et al., 2013*)). Strikingly, *gcn5Δ* and *spt20Δ* mutant cells did not show an increased longevity (27 and 17 generations, N = 50–150 cells; *Figure 11A*). Since SAGA is involved in many cellular processes, we wondered whether the observed lifespan was the result of a balance between the advantages and disadvantages of losing Gcn5 and Spt20. We rationalized that if this was the case, the longevity of these mutant cells would be limited by other factors than ERC accumulation, and thus decreasing ERC formation should not increase the longevity of these cells. Accordingly, removal of Fob1 failed to extend the lifespan of the *gcn5Δ* mutant cells (median lifespan of *gcn5Δ fob1Δ* double mutant cells = 27 generations, N = 150 cells; *Figure 11B*) and *spt20Δ* mutant cells (*spt20Δ fob1Δ* = 16 generations, N = 75 cells; *Figure 11C*). Decreased ERC accumulation in *gcn5Δ* mutant cells, as suggested by our Southern Blot experiments, and in *gcn5Δ fob1Δ* double mutant cells did not prolong lifespan of these cells, suggesting that these cells die of reasons independent of ERC accumulation. Additionally, the fact that we observed increased ERC formation

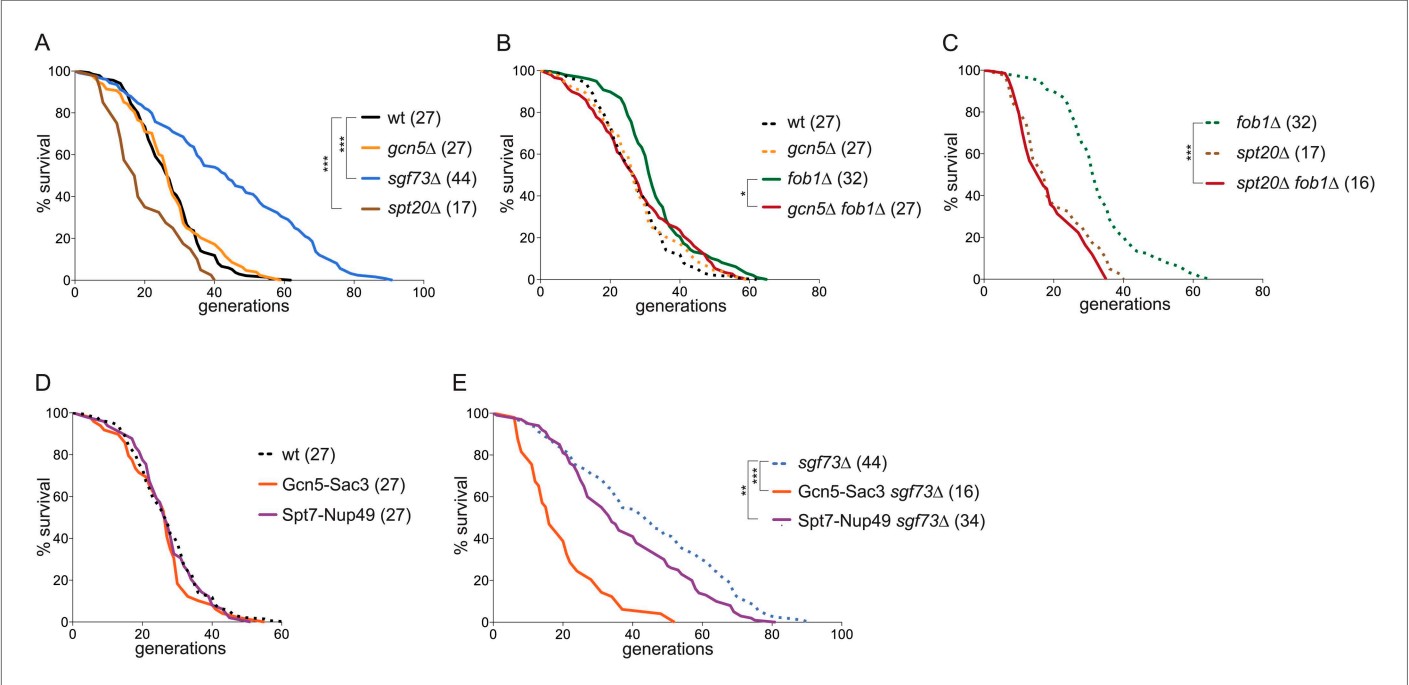

**Figure 11**. The SAGA complex promotes ageing. (**A**) Replicative lifespan (RLS) of wt, *gcn5Δ, spt20Δ*, and *sgf73Δ* cells. (**B**) RLS of *fob1Δ* and *gcn5Δ fob1Δ* cells. (**C**) RLS of *fob1Δ* and *spt20Δ fob1Δ* cells. (**D**) RLS of wt cells expressing Gcn5-Sac3 and Spt7-Nup49. (**E**) RLS of cells lacking Sgf73 and expressing Gcn5-Sac3 or Spt7-Nup49. (**A**–**D**) Median lifespan is depicted in brackets; dashed lines represent RLS of depicted wt and mutant cells of (**A**) for comparison; N = 50–150 cells; ***p < 0.001, **p < 0.01, *p < 0.05.

in cells lacking Gcn5 (*Figure 3D*) but not a shorter lifespan suggests that indeed the increased ERC formation is compensated by poor circle retention in the mother cells. Thus, cells lacking Gcn5 and Spt20 age through mechanisms independent of ERC formation and accumulation.

Finally, cells lacking Sgf73 are extremely long-lived. The fact that *sgf73Δ* mutant cells show reduced ERC formation probably contributes to this phenotype (*Figure 3D* and (*McCormick et al., 2014*)). However, the rate of ERC formation is similar in *fob1Δ* and *sgf73Δ* mutant cells, yet the *fob1Δ* mutation has a much more modest effect on lifespan (32 generations) than *sgf73Δ* (44 generations). Therefore, Sgf73 also contributes to ageing by additional mechanisms beyond reducing ERC formation. To test whether Sgf73 promotes ageing via the retention of ERCs and NPCs in the ageing cell, we asked whether targeting SAGA back to NPCs restored ageing in the *sgf73Δ* mutant cells. Expression of the fusion proteins Gcn5-Sac3 and Spt7-Nup49 did not alter the longevity of the wild-type cells (27 generations, N = 50 cells; *Figure 11D*). However, it substantially shortened the lifespan of the *sgf73Δ* mutant cells (16 and 34 generations, respectively, N ≥ 50 cells, p < 0.01; *Figure 11E*). Therefore, we conclude that decreased levels of SAGA at pores largely contribute to the extended lifespan observed in *sgf73Δ* cells. Based on the data presented here, we further conclude that SAGA promotes ageing, at least in part, through promoting the anchorage of non-chromosomal DNA circles to NPCs and promoting their joint retention and accumulation with age. Furthermore, non-chromosomal DNA circles alone do not accumulate and do not promote ageing if they are not at NPCs.

## Discussion

We show that SAGA specifies the fate of non-chromosomal circles by promoting their attachment to NPCs and thereby their high-fidelity confinement into the mother cell during mitosis. This prevents the propagation of these molecules in the population. We discovered that, as a consequence of this retention mechanism, NPCs cluster around DNA circles and are concomitantly retained in the mother cell. Thus, both ERCs and NPCs accumulate in old cells, promoting ageing. These findings allow making the following points.

## A unifying model for the asymmetric segregation of DNA circles

Our data provide a unifying model for how non-chromosomal DNA circles are retained (Model *Figure 12A*). First, we propose that both morpho-kinetic parameters (*Gehlen et al., 2011*) as well as tethering to NPCs are required for efficient circle retention in vivo. The morpho-kinetic constraints provide the baseline for circle retention and ensure that unattached plasmids are still fairly well retained, as observed in SAGA defective cells (this study) or for unattached episomes (*Gehlen et al., 2011*). Above this baseline, the barrier-tether mechanism further tightens circle retention to achieve high fidelity. Indeed, supporting this model mutations that prolong anaphase (*yku70Δ*) and mutations that weaken the nuclear diffusion barrier without prolonging anaphase (*bud6Δ*) both affect circle retention. Second, we propose that tight confinement of the circles by the diffusion barrier requires the attachment of the circles to NPCs. Indeed, what the barrier efficiently retains in the mother cell is neither NPCs alone nor the unbound circles but rather the circle-bound NPCs. Accordingly, anchorage of the circles to NPCs strongly enhances the retention of both NPCs and circles.

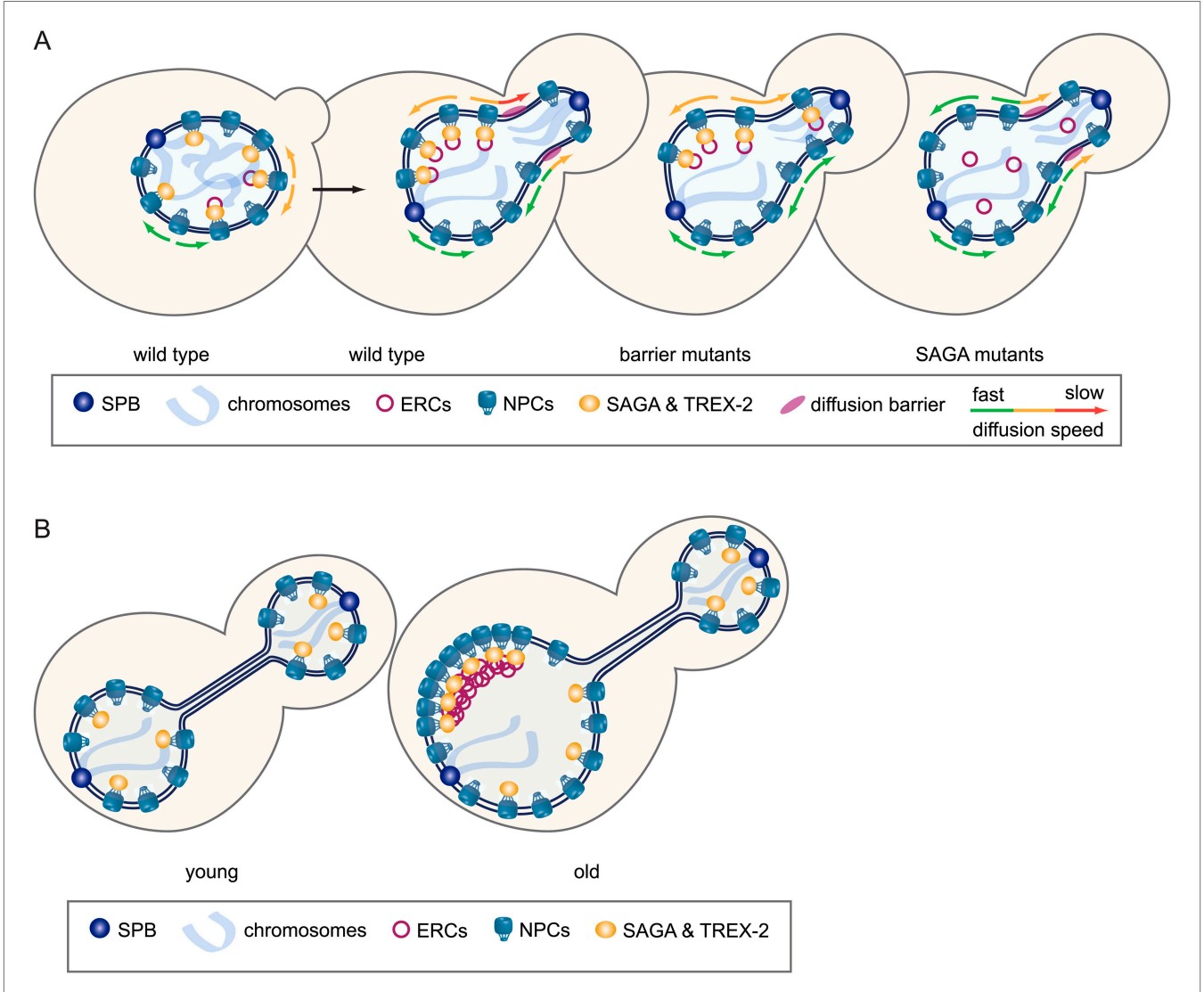

**Figure 12**. Model. (**A**) Model of how SAGA prevents DNA circles from segregating into the daughter cell: SAGA attaches DNA circles to NPCs via the TREX-2 complex, which reduces the diffusion of the circle-bound NPCs, preventing them to pass the diffusion barrier at the bud neck. (**B**) The retention of circle-bound NPCs leads to a SAGA-dependent accumulation of ERCs and NPCs in old mother cells.

This finding helps solving a debate in the field. Although NPC attachment had already been proposed to mediate the retention of DNA circles (*Shcheprova et al., 2008*), the observation that NPCs were themselves inefficiently retained in exponentially growing cells strongly argued against this model (*Khmelinskii et al., 2010*; *Colombi et al., 2013*; *Menendez-Benito et al., 2013*). Populations of exponentially growing cells are primarily composed of young cells, with only about 5% of the cells having divided more than three times. Since only very few, if any, young cells contain an ERC or other DNA circle, our model accounts well for why most exponentially growing cells only weakly retain NPCs in the mother cell.

Two types of mechanisms may underlie the selective retention of circle-bound NPCs. First, it might result from the reduced mobility of the circle-bound NPCs, which necessarily reduces the frequency at which they approach the barrier, and therefore their chance of passing it. This reduced mobility may be due to the drag that attached plasmids might oppose to NPCs. In addition, it might be exacerbated by the propensity of circle-bound NPCs to cluster, by a yet unknown mechanism. Alternatively, post-translational modifications of the NPC, perhaps by SAGA, may affect the interaction of the NPC with myosin and actin (*Steinberg et al., 2012*; *Colombi et al., 2013*) or with the lipids in the envelope, impeding their dynamics. Second, the barrier might be selective. Previous data have established that the barrier has a much larger impact on large entities than on small ones. Remodeling and clustering of the NPCs might interfere particularly strongly with their ability to pass the barrier.

Strikingly, the immobilization and segregation effects observed upon circle attachment to NPCs are reminiscent of the phenotypes caused by removing the nucleoporin Nsp1 (*Colombi et al., 2013*; *Makio et al., 2013*) or expressing defective alleles of the nucleoporins Nic96 and Nup133 (*Shcheprova et al., 2008*; *Chadrin et al., 2010*), Nic96 being a key binding partner for Nsp1 at the pore. Therefore, SAGA may regulate NPC behavior by affecting specific parts of the pore itself.

Together, our observations explain why the diffusion barrier in the envelope of anaphase nuclei, which has limited effects on bulk NPC distribution, has a crucial impact on ageing. Indeed, ageing cells accumulate circles with increasing speed. When the circles cannot attach or when the diffusion barrier is defective, the morpho-kinetic model still promotes a retention level of about 88–85%. However, our data indicate that the loss of 12–15% of the circles per division cycle has a substantial impact on ERC accumulation and the longevity of the cells, the *bud6Δ* and the *sgf73Δ* mutant cells being remarkably long-lived. This large impact on longevity for apparently small defects in retention is fully in line with the idea that high-fidelity retention is a crucial parameter, as predicted by in silico simulations (*Gillespie et al., 2004*; *Shcheprova et al., 2008*).

Our data indicate that, among its many functions, the role of SAGA in linking chromatin to NPCs is especially crucial for circle retention and ageing. Thus, while the gene-gating hypothesis designed to account for SAGA-dependent transcription at NPCs is debated (*Cabal et al., 2006*; *Green et al., 2012*), our findings may provide an alternative model for SAGA's function at the pore.

How SAGA promotes circle attachment to NPCs is not fully understood at this point. SAGA might modify chromatin or some chromatin-binding component of the NPC and thereby regulate the interaction of specific DNA pieces with the pore. Alternatively, it may function itself as a physical tether. Although our data do not allow any definitive conclusion, it is remarkable that SAGA needs to be both on the NPC and on the chromatin to mediate its function in circle attachment to pores. Thus, we favor the idea that SAGA is an integral part of the tether. The fact that its acetyl transferase activity is also needed for its function might reflect a dual role of SAGA, both as a tether and as a regulator of tethering. Identifying the acetylation targets of SAGA and the role of these modifications in circle retention are now required in order to address how SAGA facilitates chromatin anchorage to NPCs.

## SAGA helps to discriminate self from non-self DNA

One intriguing aspect of DNA circle retention is that it affects a very large spectrum of molecules. Together with the current knowledge, the identification of SAGA as a player in this process suggests an explanation for this observation. The SAGA complex, which associates broadly with chromatin and interacts with every expressed sequence (*Ohtsuki et al., 2010*; *Bonnet et al., 2014*), promotes DNA anchorage to NPCs (*Luthra et al., 2007*). Current data suggest that this association occurs on both chromosomal and non-chromosomal DNA. Nevertheless, we observed an enrichment of SAGA on non-centromeric DNA circles both by using microscopy and ChIP experiments. Therefore, we propose that either SAGA is more efficiently recruited to or more stably associated with acentric DNA molecules or, alternatively, that during mitosis SAGA interaction or function is specifically repressed on chromosomal DNA.

Strikingly, such a mechanism has the potential of discriminating self from foreign DNA. Since circles containing transcribed sequences are expected to recruit more SAGA, they may be more efficiently marked and retained than non-transcribed circles. In addition, our ChIP data are consistent with previous data indicating that ARSs are also SAGA targets (*Espinosa et al., 2010*). In fitting, DNA circles with strong replication origins are better retained in mother cells than those with weak origins (*Futcher and Cox, 1984*). Thus, replicating circles may also be better retained than non-replicating circles. Therefore, a SAGA-based mechanism to discriminate between chromosomal and non-chromosomal DNA would primarily target non-chromosomal DNA molecules of internal or foreign origin based on their expression and replication potential, i.e. on their potential toxicity. Hence, we suggest that this pathway fulfills the functions of a genomic immunity system capable of preventing the spread of extra-chromosomal DNA molecules of both foreign and endogenous origin in a population. Since DNA circles form in all organisms tested so far (*Gaubatz, 1990*; *Cohen et al., 2003*, *2010*), there must exist some mechanism to restrict their propagation in higher eukaryotes as well.

### DNA circles and ageing

Together, our data suggest a mechanism for the effect of non-chromosomal DNA circles on cellular longevity. This mechanism relies not on the presence of the circles themselves but on their SAGA-dependent association with NPCs (Model *Figure 12B*). Therefore, replicative ageing in yeast might be at least in part the secondary consequence of a mechanism involved in restricting the propagation of non-chromosomal DNA, but inevitably altering nuclear organization and NPC number.

Human diseases leading to premature ageing phenotypes, such as Bloom and Werner syndromes, originate from dysfunctions in two RecQ helicases involved in DNA recombination, BLM, and WRN (*Burtner and Kennedy, 2010*). RecQ helicases inhibit the formation of Holliday junctions (*Sharma et al., 2006*) and therefore of circles. Thus, an increased rate of circle formation might underlie at least some aspects of the Werner and Bloom syndromes. Interestingly, the Hutchinson–Gilford progeria syndrome (HGPS), which is caused by the defects of the nuclear lamina and causes alterations of the nuclear structure (*Scaffidi and Misteli, 2006*), leads to very similar ageing phenotypes as Werner and Bloom syndromes. These different observations indicate a link between genomic instability, DNA circle formation, and changes in nuclear organization during ageing and suggest that NPCs might be at the center of these relationships. Together, our data indicate that yeast is a powerful system to dissect how NPC number and modifications affect nuclear homeostasis and cellular viability.

## Materials and methods

### Strains and plasmids

Yeast strains used in this study are listed in *Table 1*. All strains are isogenic to S288C, except the strain containing the *ADE2* gene in the rDNA locus to check the rate of ERC formation, which is derived from W303 and was a gift from Lorraine Pillus (*Jacobson and Pillus, 2009*). GFP and knock out strains derived from collections (http://web.uni-frankfurt.de/fb15/mikro/euroscarf/index.html and http://clones.lifetechnologies.com/cloneinfo.php?clone=yeastgfp) or were manually created as described (*Janke et al., 2004*). Cells were grown using the standard conditions: YPD at 30°C unless otherwise indicated. pYB1601 containing 3sfGFP was created by Mathias Bayer. pDS316 containing one rDNA repeat was a gift from D. A. Sinclair (*Sinclair and Guarente, 1997*) and pYB1382 containing *gcn5-E173A* was a gift from S. L. Berger (*Wang et al., 1998*). *gcn5-E173A* was integrated into the genome to replace *GCN5*. To allow the visualization of SPBs and the expression of the recombinase upon addition of estradiol, *spc42::SPC42-CFP:kanMX4* and the estradiol binding domain fused to Gal4 *trp1::GAL4-EBD:TRP1* were introduced into the plasmid visualization strain used in *Shcheprova et al. (2008)* . To create TetR fusion proteins, a plasmid pYB1666 containing *linker-TETR-mCHERRY-hphNT1* was created by homologous recombination into pRS314. PCR products amplifying *linker-TETR-mCHERRY-hphNT1* and containing homologous overhangs were used to tag endogenous genes with TetR and correct integrations were checked by microscopy. To generate the Gcn5-Sac3 and Spt7-Nup49 fusion proteins, pYB1665 containing *GCN5-linker-SAC3-SAC3(3′UTR)-hphNT1-GCN5(3′UTR)* and pYB1739 containing *SPT7-linker-NUP49-NUP49(3′UTR)hphNT1-SPT7(3′UTR)* were created by homologous recombination into pRS314. PCR fragments containing the fusions were treated with DpnI and gel-purified before integration into yeast. Integrations were verified by PCR. For all fusion proteins the linker sequence is as described in *Gruber et al. (2006)*.

**Table 1.** Strain list

| yYB number | | Genotype |
|---|---|---|
| 6637 | a | *nup82::NUP82-3sfGFP:kanMX4; pPCM14 (224 tetO-REC-URA3-CEN-REC-LEU2); P_URA3-TETR-mCherry:kanMX4; his3::P_GAL-REC:HIS3; trp1::GAL4-EBD:TRP1; ade2-101* |
| 10,071, 10,072, 10,073 | a | *fob1::FOB1-yeGFP:hphNT1; pPCM14 (224 tetO-REC-URA3-CEN-REC-LEU2); P_URA3-TETR-mCherry:kanMX4; his3::P_GAL-REC:HIS3; trp1::GAL4-EBD:TRP1; ade2-101; clones 1-3* |
| 6271 | a | *nup82::NUP82-yeGFP:natNT2; pPCM14 (224 tetO-REC-URA3-CEN-REC-LEU2); P_URA3-TETR-mCherry:kanMX4; his3::P_GAL-REC:HIS3; trp1::GAL4-EBD:TRP1; ade2-101* |
| 9038 | a | *nup49::NUP49-yeGFP:hphNT1; pPCM14 (224 tetO-REC-URA3-CEN-REC-LEU2); P_URA3-TETR-mCherry:kanMX4; his3::P_GAL-REC:HIS3; trp1::GAL4-EBD:TRP1; ade2-101* |
| 9415, 9416, 10,454 | a | *nup49::NUP49-yeGFP:hphNT1; bud6::natNT2; pPCM14 (224 tetO-REC-URA3-CEN-REC-LEU2); P_URA3-TETR-mCherry:kanMX4; his3::P_GAL-REC:HIS3; trp1::GAL4-EBD:TRP1; ade2-101 clones 1-3* |
| 4221 | a | *pPCM14 (224 tetO-REC-URA3-CEN-REC-LEU2); spc42::SPC42-CFP:kanMX4; leu2::TETR-GFP:LEU2; his3::P_GAL-REC:HIS3; trp1::GAL4-EBD:TRP1; ade2-101* |
| 4222, 5547, 6521 | a | *bud6::natNT2; pPCM14 (224 tetO-REC-URA3-CEN-REC-LEU2); spc42::SPC42-CFP:kanMX4; leu2::TETR-GFP:LEU2; his3::P_GAL-REC:HIS3; trp1::GAL4-EBD:TRP1; ade2-101; clones 1-3* |
| 5516, 5517, 8099 | a | *yku70::hphNT1; pPCM14 (224 tetO-REC-URA3-CEN-REC-LEU2); spc42::SPC42-CFP:kanMX4; leu2::TETR-GFP:LEU2; his3::P_GAL-REC:HIS3; trp1::GAL4-EBD:TRP1; ade2-101; clones 1-3* |
| 4124, 4161, 7237 | a | *sir1::natNT2; pPCM14 (224 tetO-REC-URA3-CEN-REC-LEU2); spc42::SPC42-CFP:kanMX4; leu2::TETR-GFP:LEU2; his3::P_GAL-REC:HIS3; trp1::GAL4-EBD:TRP1; ade2-101; clones 1-3* |
| 4165, 4260, 4261 | a | *sir2::natNT2; pPCM14 (224 tetO-REC-URA3-CEN-REC-LEU2); spc42::SPC42-CFP:kanMX4; leu2::TETR-GFP:LEU2; his3::P_GAL-REC:HIS3; trp1::GAL4-EBD:TRP1; ade2-101; clones 1-3* |
| 4121, 7239, 7240 | a | *sir3::natNT2; pPCM14 (224 tetO-REC-URA3-CEN-REC-LEU2); spc42::SPC42-CFP:kanMX4; leu2::TETR-GFP:LEU2; his3::P_GAL-REC:HIS3; trp1::GAL4-EBD:TRP1; ade2-101; clones 1-3* |
| 4235, 4236, 7243 | a | *sir4::natNT2; pPCM14 (224 tetO-REC-URA3-CEN-REC-LEU2); spc42::SPC42-CFP:kanMX4; leu2::TETR-GFP:LEU2; his3::P_GAL-REC:HIS3; trp1::GAL4-EBD:TRP1; ade2-101; clones 1-3* |
| 8100, 8101, 8102 | a | *mps3::mps3-aa75-100Δ:hphNT1; pPCM14 (224 tetO-REC-URA3-CEN-REC-LEU2); spc42::SPC42-CFP:kanMX4; leu2::TETR-GFP:LEU2; his3::P_GAL-REC:HIS3; trp1::GAL4-EBD:TRP1; ade2-101; clones 1-3* |
| 6524, 6652, 8059 | a | *src1::natNT2; pPCM14 (224 tetO-REC-URA3-CEN-REC-LEU2); spc42::SPC42-CFP:kanMX4; leu2::TETR-GFP:LEU2; his3::P_GAL-REC:HIS3; trp1::GAL4-EBD:TRP1; ade2-101; clones 1-3* |
| 7957, 7958, 7960 | a | *heh2::hphNT1; pPCM14 (224 tetO-REC-URA3-CEN-REC-LEU2); spc42::SPC42-CFP:kanMX4; leu2::TETR-GFP:LEU2; his3::P_GAL-REC:HIS3; trp1::GAL4-EBD:TRP1; ade2-101; clones 1-3* |
| 7954, 7955, 7956 | a | *slx5::hphNT1; pPCM14 (224 tetO-REC-URA3-CEN-REC-LEU2); spc42::SPC42-CFP:kanMX4; leu2::TETR-GFP:LEU2; his3::P_GAL-REC:HIS3; trp1::GAL4-EBD:TRP1; ade2-101; clones 1-3* |
| 4120, 4263, 4264 | a | *gcn5::natNT2; pPCM14 (224 tetO-REC-URA3-CEN-REC-LEU2); spc42::SPC42-CFP:kanMX4; leu2::TETR-GFP:LEU2; his3::P_GAL-REC:HIS3; trp1::GAL4-EBD:TRP1; ade2-101; clones 1-3* |
| 4118, 4119, 4262 | a | *spt3::natNT2; pPCM14 (224 tetO-REC-URA3-CEN-REC-LEU2); spc42::SPC42-CFP:kanMX4; leu2::TETR-GFP:LEU2; his3::P_GAL-REC:HIS3; trp1::GAL4-EBD:TRP1; ade2-101; clones 1-3* |
| 4287, 4577, 5518 | a | *sgf73::natNT2; pPCM14 (224 tetO-REC-URA3-CEN-REC-LEU2); spc42::SPC42-CFP:kanMX4; leu2::TETR-GFP:LEU2; his3::P_GAL-REC:HIS3; trp1::GAL4-EBD:TRP1; ade2-101; clones 1-3* |
| 5319, 5320, 5321 | a | *gcn5::gcn5E173A:hphNT1; pPCM14 (224 tetO-REC-URA3-CEN-REC-LEU2); spc42::SPC42-CFP:kanMX4; leu2::TETR-GFP:LEU2; his3::P_GAL-REC:HIS3; trp1::GAL4-EBD:TRP1; ade2-101; clones 1-3* |

*Table 1. Continued on next page*

*Table 1. Continued*

| yYB number | | Genotype |
|---|---|---|
| 4166, 4237, 4238 | a | *sac3::natNT2; pPCM14 (224 tetO-REC-URA3-CEN-REC-LEU2); spc42::SPC42-CFP: kanMX4; leu2::TETR-GFP:LEU2; his3::P$_{GAL}$-REC:HIS3; trp1::GAL4-EBD:TRP1; ade2-101;* clones 1-3 |
| 4122, 4573, 8103 | a | *sus1::natNT2; pPCM14 (224 tetO-REC-URA3-CEN-REC-LEU2); spc42::SPC42-CFP: kanMX4; leu2::TETR-GFP:LEU2; his3::P$_{GAL}$-REC:HIS3; trp1::GAL4-EBD:TRP1; ade2-101;* clones 1-3 |
| 384 | a | *ura3-52; his3Δ200; leu2; lys2-801; ade2-101; trp1Δ63* |
| 3870 | a | *gcn5::natNT2; ura3-52; his3Δ200; leu2; lys2-801; ade2-101; trp1Δ63* |
| 4287 | a | *sgf73::natNT2; ura3-52; his3Δ200; leu2; lys2-801; ade2-101; trp1Δ63* |
| 3415 | a | *hoD::P$_{SCW11}$-Cre-EBD78:natMX; ubc9::loxP-UBC9-loxP:LEU2; cdc20::loxP-CDC20-intron-loxP:hphMX; ade2::hisG; his3; lys2; ura3; trp1Δ63* |
| 6580 | a | *gcn5::kanMX4; hoD::P$_{SCW11}$-Cre-EBD78:natMX; ubc9::loxP-UBC9-loxP:LEU2; cdc20::loxP-CDC20-intron-loxP:hphMX; ade2::hisG; his3; lys2; ura3; trp1Δ63* |
| 6195 | a | *sgf73::kanMX4; hoD::P$_{SCW11}$-Cre-EBD78:natMX; ubc9::loxP-UBC9-loxP:LEU2; cdc20::loxP-CDC20-intron-loxP:hphMX; ade2::hisG; his3; lys2; ura3; trp1Δ63* |
| 5457 | a | *pPCM14 (224 tetO-REC-URA3-CEN-REC-LEU2); spc42::SPC42-redStar:natNT2; nup170::NUP170-mCherry:kanMX4; leu2::TETR-GFP:LEU2; his3::P$_{GAL}$-REC:HIS3; trp1::GAL4-EBD:TRP1; ade2-101;* |
| 5502, 5503, 5504 | a | *gcn5::hphNT1; pPCM14 (224 tetO-REC-URA3-CEN-REC-LEU2); spc42::SPC42-redStar:natNT2; nup170::NUP170-mCherry:kanMX4; leu2::TETR-GFP:LEU2; his3::P$_{GAL}$-REC:HIS3; trp1::GAL4-EBD:TRP1; ade2-101;* clones 1-3 |
| 8115, 8116, 8117 | a | *sus1::hphNT1; pPCM14 (224 tetO-REC-URA3-CEN-REC-LEU2); spc42::SPC42-redStar:natNT2; nup170::NUP170-mCherry:kanMX4; leu2::TETR-GFP:LEU2; his3::P$_{GAL}$-REC:HIS3; trp1::GAL4-EBD:TRP1; ade2-101;* clones 1-3 |
| 6648 | a | *pPCM14 (224 tetO-REC-URA3-CEN-REC-LEU2); nup82::NUP82-3sfGFP:kanMX4; P$_{URA3}$-TETR-mCherry:kanMX4; spc42::SPC42-yeGFP:hphNT1; his3::P$_{GAL}$-REC:HIS3; trp1::GAL4-EBD:TRP1; ade2-101* |
| 6752, 6765, 6895 | a | *gcn5::natNT2; pPCM14 (224 tetO-REC-URA3-CEN-REC-LEU2); nup82::NUP82-3sfGFP:kanMX4; P$_{URA3}$-TETR-mCherry:kanMX4; spc42::SPC42-yeGFP:hphNT1; his3::P$_{GAL}$-REC:HIS3; trp1::GAL4-EBD:TRP1; ade2-101;* clones 1-3 |
| 8093, 8094, 8095 | a | *sus1::natNT2; pPCM14 (224 tetO-REC-URA3-CEN-REC-LEU2); nup82::NUP82-3sfGFP:kanMX4; P$_{URA3}$-TETR-mCherry:kanMX4; spc42::SPC42-yeGFP:hphNT1; his3::P$_{GAL}$-REC:HIS3; trp1::GAL4-EBD:TRP1; ade2-101;* clones 1-3 |
| 5881, 5882, 5883 | a | *nup170::NUP170-linker-TETR-mCherry:hphNT1; pPCM14 (224 tetO-REC-URA3-CEN-REC-LEU2); spc42::SPC42-CFP:kanMX4; leu2::TETR-GFP:LEU2; his3::P$_{GAL}$-REC:HIS3; trp1::GAL4-EBD:TRP1; ade2-101;* clones 1-3 |
| 5886, 7253, 7254 | a | *nup170::NUP170-linker-TETR-mCherry:hphNT1; gcn5::natNT2; pPCM14 (224 tetO-REC-URA3-CEN-REC-LEU2); spc42::SPC42-CFP:kanMX4; leu2::TETR-GFP:LEU2; his3::P$_{GAL}$-REC:HIS3; trp1::GAL4-EBD:TRP1; ade2-101;* clones 1-3 |
| 5962, 5973, 7863 | a | *nup170::NUP170-linker-TETR-mCherry:hphNT1; sus1::natNT2; pPCM14 (224 tetO-REC-URA3-CEN-REC-LEU2); spc42::SPC42-CFP:kanMX4; leu2::TETR-GFP:LEU2; his3::P$_{GAL}$-REC:HIS3; trp1::GAL4-EBD:TRP1; ade2-101;* clones 1-3 |
| 5933, 5934, 5935 | a | *nup49::NUP49-linker-TETR-mCherry:hphNT1; pPCM14 (224 tetO-REC-URA3-CEN-REC-LEU2); spc42::SPC42-CFP:kanMX4; leu2::TETR-GFP:LEU2; his3::P$_{GAL}$-REC:HIS3; trp1::GAL4-EBD:TRP1; ade2-101;* clones 1-3 |
| 7959, 8032, 8033 | a | *nup49::NUP49-linker-TETR-mCherry:hphNT1; gcn5::natNT2; pPCM14 (224 tetO-REC-URA3-CEN-REC-LEU2); spc42::SPC42-CFP:kanMX4; leu2::TETR-GFP:LEU2; his3::P$_{GAL}$-REC:HIS3; trp1::GAL4-EBD:TRP1; ade2-101;* clones 1-3 |
| 7865, 7866, 7867 | a | *nup49::NUP49-linker-TETR-mCherry:hphNT1; sgf73::natNT2; pPCM14 (224 tetO-REC-URA3-CEN-REC-LEU2); spc42::SPC42-CFP:kanMX4; leu2::TETR-GFP:LEU2; his3::P$_{GAL}$-REC:HIS3; trp1::GAL4-EBD:TRP1; ade2-101;* clones 1-3 |
| 5884, 5885, 5961 | a | *nup170::NUP170-linker-TETR-mCherry:hphNT1; sgf73::natNT2; pPCM14 (224 tetO-REC-URA3-CEN-REC-LEU2); spc42::SPC42-CFP:kanMX4; leu2::TETR-GFP:LEU2; his3::P$_{GAL}$-REC:HIS3; trp1::GAL4-EBD:TRP1; ade2-101;* clones 1-3 |
| 5966, 5967, 7864 | a | *nup49::NUP49-linker-TETR-mCherry:hphNT1; sus1::natNT2; pPCM14 (224 tetO-REC-URA3-CEN-REC-LEU2); spc42::SPC42-CFP:kanMX4; leu2::TETR-GFP:LEU2; his3::P$_{GAL}$-REC:HIS3; trp1::GAL4-EBD:TRP1; ade2-101;* clones 1-3 |

Table 1. Continued

| yYB number | | Genotype |
|---|---|---|
| 5968, 5969, 5970 | a | nup49::NUP49-linker-TETR-mCherry:hphNT1; bud6::natNT2; pPCM14 (224 tetO-REC-URA3-CEN-REC-LEU2); spc42::SPC42-CFP:kanMX4; leu2::TETR-GFP:LEU2; his3::$P_{GAL}$-REC:HIS3; trp1::GAL4-EBD:TRP1; ade2-101; clones 1-3 |
| 5963, 5964, 5965 | a | nup170::NUP170-linker-TETR-mCherry:hphNT1; bud6::natNT2; pPCM14 (224 tetO-REC-URA3-CEN-REC-LEU2); spc42::SPC42-CFP:kanMX4; leu2::TETR-GFP:LEU2; his3::$P_{GAL}$-REC:HIS3; trp1::GAL4-EBD:TRP1; ade2-101; clones 1-3 |
| 5852, 5853,5854 | a | gcn5::GCN5-linker-SAC3:hphNT1; pPCM14 (224 tetO-REC-URA3-CEN-REC-LEU2); spc42::SPC42-CFP:kanMX4; leu2::TETR-GFP:LEU2; his3::$P_{GAL}$-REC:HIS3; trp1::GAL4-EBD:TRP1; ade2-101; clones 1-3 |
| 7141, 7142, 7143 | a | spt7::SPT7-linker-NUP49:hphNT1; pPCM14 (224 tetO-REC-URA3-CEN-REC-LEU2); spc42::SPC42-CFP:kanMX4; leu2::TETR-GFP:LEU2; his3::$P_{GAL}$-REC:HIS3; trp1::GAL4-EBD:TRP1; ade2-101; clones 1-3 |
| 5888, 5889, 7856 | a | gcn5::GCN5-linker-SAC3:hphNT1; sus1::natNT2; pPCM14 (224 tetO-REC-URA3-CEN-REC-LEU2); spc42::SPC42-CFP:kanMX4; leu2::TETR-GFP:LEU2; his3::$P_{GAL}$-REC:HIS3; trp1::GAL4-EBD:TRP1; ade2-101; clones 1-3 |
| 7859, 7869, 7870 | a | spt7::SPT7-linker-NUP49:hphNT1; sus1::natNT2; pPCM14 (224 tetO-REC-URA3-CEN-REC-LEU2); spc42::SPC42-CFP:kanMX4; leu2::TETR-GFP:LEU2; his3::$P_{GAL}$-REC:HIS3; trp1::GAL4-EBD:TRP1; ade2-101; clones 1-3 |
| 6048, 7858, 7868 | a | gcn5::GCN5-linker-SAC3:hphNT1; sgf73::natNT2; pPCM14 (224 tetO-REC-URA3-CEN-REC-LEU2); spc42::SPC42-CFP:kanMX4; leu2::TETR-GFP:LEU2; his3::$P_{GAL}$-REC:HIS3; trp1::GAL4-EBD:TRP1; ade2-101; clones 1-3 |
| 7557, 7860, 7861 | a | spt7::SPT7-linker-NUP49:hphNT1; sgf73::natNT2; pPCM14 (224 tetO-REC-URA3-CEN-REC-LEU2); spc42::SPC42-CFP:kanMX4; leu2::TETR-GFP:LEU2; his3::$P_{GAL}$-REC:HIS3; trp1::GAL4-EBD:TRP1; ade2-101; clones 1-3 |
| 9520, 9521, 9522 | a | gcn5::GCN5-yeGFP:hphNT1; pPCM14 (224 tetO-REC-URA3-CEN-REC-LEU2); $P_{URA3}$-TETR-mCherry:kanMX4; his3::$P_{GAL}$-REC:HIS3; trp1::GAL4-EBD:TRP1; ade2-101; clones 1-3 |
| 9523, 9524, 9525 | a | sgf73::SGF73-yeGFP:hphNT1; pPCM14 (224 tetO-REC-URA3-CEN-REC-LEU2); $P_{URA3}$-TETR-mCherry:kanMX4; his3::$P_{GAL}$-REC:HIS3; trp1::GAL4-EBD:TRP1; ade2-101; clones 1-3 |
| 7311 | α | pYB1670 (REC-URA3-CEN-REC-ARS1-LEU2-AmpR); his3::$P_{GAL}$-REC:HIS3; trp1::GAL4-EBD:TRP1; leu2Δ0; ura3Δ0; met15Δ0 |
| 7555 | a | gcn5::GCN5-yeGFP:HIS3; pYB1670 (REC-URA3-CEN-REC-ARS1-LEU2-AmpR); his3::$P_{GAL}$-REC:HIS3; trp1::GAL4-EBD:TRP1; leu2Δ0; ura3Δ0; met15Δ0 |
| 7524 | α | spt20::SPT20-yeGFP:HIS3; pYB1670 (REC-URA3-CEN-REC-ARS1-LEU2-AmpR); his3::$P_{GAL}$-REC:HIS3; trp1::GAL4-EBD:TRP1; leu2Δ0; ura3Δ0; met15Δ0 |
| 8831, 8832, 8833 | a | nup82::NUP82-3sfGFP:kanMX4; gcn5::hphNT1; pPCM14 (224 tetO-REC-URA3-CEN-REC-LEU2); $P_{URA3}$-TETR-mCherry:kanMX4; his3::$P_{GAL}$-REC:HIS3; trp1::GAL4-EBD:TRP1; ade2-101; clones 1-3 |
| 9002, 9003, 10,455 | a | nup82::NUP82-3sfGFP:kanMX4; sgf73::hphNT1; pPCM14 (224 tetO-REC-URA3-CEN-REC-LEU2); $P_{URA3}$-TETR-mCherry:kanMX4; his3::$P_{GAL}$-REC:HIS3; trp1::GAL4-EBD:TRP1; ade2-101; clones 1&2 |
| 9039 | a | nup170::NUP170-yeGFP:hphNT1; pPCM14 (224 tetO-REC-URA3-CEN-REC-LEU2); $P_{URA3}$-TETR-mCherry:kanMX4; his3::$P_{GAL}$-REC:HIS3; trp1::GAL4-EBD:TRP1; ade2-101; clones 1-3 |
| 9392 | a | nup170::NUP170-yeGFP:hphNT1; gcn5::natNT2; pPCM14 (224 tetO-REC-URA3-CEN-REC-LEU2); $P_{URA3}$-TETR-mCherry:kanMX4; his3::$P_{GAL}$-REC:HIS3; trp1::GAL4-EBD:TRP1; ade2-101 |
| 7828 | a | nup49::NUP49-yeGFP:HIS3; leu2Δ0; ura3Δ0; met15Δ0 |
| 8112 | a | nup170::NUP170-yeGFP:HIS3; leu2Δ0; ura3Δ0; met15Δ0 |
| 8182 | a | nup82::NUP82-yeGFP:HIS3; leu2Δ0; ura3Δ0; met15Δ0 |
| 9337, 9338, 9339 | a | nup170::NUP170-yeGFP:HIS3; bud6::kanMX4; leu2Δ0; ura3Δ0; met15Δ0; clones 1-3 |
| 9612, 9613, 9614 | a | nup170::NUP170-yeGFP:HIS3; fob1::kanMX4; leu2Δ0; ura3Δ0; met15Δ0; clones 1-3 |
| 9616, 9617, 9618 | a | nup170::NUP170-yeGFP:HIS3; sir2::kanMX4; leu2Δ0; ura3Δ0; met15Δ0; clones 1-3 |
| 9372, 9373, 9374 | a | nup170::NUP170-yeGFP:HIS3; sgf73::hphNT1; leu2Δ0; ura3Δ0; met15Δ0; clones 1-3 |

Table 1. Continued on next page

*Table 1. Continued*

| yYB number | | Genotype |
|---|---|---|
| 9171, 9173, 9174 | a | *nup49::NUP49-yeGFP:HIS3; fob1::kanMX4; leu2Δ0; ura3Δ0; met15Δ0;* clones 1-3 |
| 9181, 9182, 9183 | a | *nup49::NUP49-yeGFP:HIS3; sir2::kanMX4; leu2Δ0; ura3Δ0; met15Δ0;* clones 1-3 |
| 4499, 4500, 4501 | a | *nup49::NUP49-yeGFP:HIS3; sgf73::kanMX4; leu2Δ0; ura3Δ0; met15Δ0;* clones 1-3 |
| 9360, 9361, 9362 | a | *nup170::NUP170-yeGFP:HIS3; sgf73::kanMX4; spt7::SPT7-linker-NUP49:hphNT1; leu2Δ0; ura3Δ0; met15Δ0;* clones 1-3 |
| 9340, 9341, 9342 | a | *nup170::NUP170-yeGFP:HIS3; sgf73::kanMX4; gcn5::GCN5-linker-SAC3:hphNT1; leu2Δ0; ura3Δ0; met15Δ0;* clones 1-3 |
| 5520 | a | *his3Δ1; leu2Δ0; ura3Δ0; met15Δ0 (EUROSCARF wt)* |
| 5142 | a | *gcn5::kanMX4; his3Δ1; leu2Δ0; ura3Δ0; met15Δ0* |
| 8108 | a | *sgf73::kanMX4; his3Δ1; leu2Δ0; ura3Δ0; met15Δ0* |
| 10,453 | a | *fob1::kanMX4; his3Δ1; leu2Δ0; ura3Δ0; met15Δ0* |
| 9915, 9916, 9917 | a | *fob1::kanMX4; gcn5::hphNT1; his3Δ1; leu2Δ0; ura3Δ0; met15Δ0;* clones 1-3 |
| 6626 | a | *gcn5::GCN5-linker-SAC3:hphNT1; his3Δ1; leu2Δ0; ura3Δ0; met15Δ0* |
| 7252 | a | *spt7::SPT7-linker-NUP49:hphNT1; his3Δ1; leu2Δ0; ura3Δ0; met15Δ0* |
| 6632, 6633, 6634 | a | *gcn5::GCN5-linker-SAC3:hphNT1; sgf73::kanMX4; his3Δ1; leu2Δ0; ura3Δ0; met15Δ0;* clones 1-3 |
| 7540, 7541, 7542 | a | *spt7::SPT7-linker-NUP49:hphNT1; sgf73::kanMX4; his3Δ1; leu2Δ0; ura3Δ0; met15Δ0;* clones 1-3 |
| 5532 | a | *nsg1::NSG1-yeGFP:HIS3; leu2Δ0; ura3Δ0; met15Δ0* |
| 4233, 7303, 7304 | a | *bud6::natNT2; nsg1::NSG1-yeGFP:HIS3; leu2Δ0; ura3Δ0; met15Δ0;* clones 1-3 |
| 6897, 8105, 8106 | a | *yku70::hphNT1; nsg1::NSG1-yeGFP:HIS3; leu2Δ0; ura3Δ0; met15Δ0;* clones 1-3 |
| 6766, 6767, 8104 | a | *gcn5::natNT2; nsg1::NSG1-yeGFP:HIS3; leu2Δ0; ura3Δ0; met15Δ0;* clones 1-3 |
| 4601, 4615, 4616 | a | *sgf73::kanMX4; nsg1::NSG1-yeGFP:HIS3; leu2Δ0; ura3Δ0; met15Δ0;* clones 1-3 |
| 2177 | α | *nup49::NUP49-yeGFP:HIS3; leu2Δ0; ura3Δ0; met15Δ0* |
| 3403, 3583, 4223 | α | *bud6::kanMX4; nup49::NUP49-yeGFP:HIS3; leu2Δ0; ura3Δ0; met15Δ0;* clones 1-3 |
| 4094, 4095, 4096 | α | *gcn5::natNT2; nup49::NUP49-yeGFP:HIS3; leu2Δ0; ura3Δ0; met15Δ0;* clones 1-3 |
| 8109 | a | *nup2::NUP2-yeGFP:HIS3; leu2Δ0; ura3Δ0; met15Δ0* |
| 8110 | a | *nup60::NUP60-yeGFP:HIS3; leu2Δ0; ura3Δ0; met15Δ0* |
| 8111 | a | *mlp1::MLP1-yeGFP:HIS3; leu2Δ0; ura3Δ0; met15Δ0* |
| 8112 | a | *nup170::NUP170-yeGFP:HIS3; leu2Δ0; ura3Δ0; met15Δ0* |
| 7828 | a | *nup49::NUP49-yeGFP:HIS3; leu2Δ0; ura3Δ0; met15Δ0* |
| 7336 | a | *nup1::NUP1-yeGFP:HIS3; leu2Δ0; ura3Δ0; met15Δ0* |
| 7503, 8113, 8114 | a | *nup1::NUP1-linker-TETR-mCherry:hphNT1; pPCM14 (224 tetO-REC-URA3-CEN-REC-LEU2); spc42::SPC42-CFP:kanMX4; leu2::TETR-GFP:LEU2; his3::P$_{GAL}$-REC:HIS3; trp1::GAL4-EBD:TRP1; ade2-101;* clones 1-3 |
| 5933, 5934, 5935 | a | *nup49::NUP49-linker-TETR-mCherry:hphNT1; pPCM14 (224 tetO-REC-URA3-CEN-REC-LEU2); spc42::SPC42-CFP:kanMX4; leu2::TETR-GFP:LEU2; his3::P$_{GAL}$-REC:HIS3; trp1::GAL4-EBD:TRP1; ade2-101;* clones 1-3 |
| 5881, 5882, 5883 | a | *nup170::NUP170-linker-TETR-mCherry:hphNT1; pPCM14 (224 tetO-REC-URA3-CEN-REC-LEU2); spc42::SPC42-CFP:kanMX4; leu2::TETR-GFP:LEU2; his3::P$_{GAL}$-REC:HIS3; trp1::GAL4-EBD:TRP1; ade2-101;* clones 1-3 |
| 5930, 5931, 5932 | a | *mlp1::MLP1-linker-TETR-mCherry:hphNT1; pPCM14 (224 tetO-REC-URA3-CEN-REC-LEU2); spc42::SPC42-CFP:kanMX4; leu2::TETR-GFP:LEU2; his3::P$_{GAL}$-REC:HIS3; trp1::GAL4-EBD:TRP1; ade2-101;* clones 1-3 |
| 5843, 5844, 5845 | a | *nup60::NUP60-linker-TETR-mCherry:hphNT1; pPCM14 (224 tetO-REC-URA3-CEN-REC-LEU2); spc42::SPC42-CFP:kanMX4; leu2::TETR-GFP:LEU2; his3::P$_{GAL}$-REC:HIS3; trp1::GAL4-EBD:TRP1; ade2-101;* clones 1-3 |
| 10,789 | a | *spt20::kanMX4; his3Δ1; leu2Δ0; ura3Δ0; met15Δ0* |
| 10,804, 10,805, 10,806 | a | *spt20::kanMX4; fob1::hphNT1 his3Δ1; leu2Δ0; ura3Δ0; met15Δ0;*clones 1-3 |

*Table 1. Continued on next page*

*Table 1. Continued*

| yYB number | | Genotype |
|---|---|---|
| 11,120 | a | *rDNA::ADE2-CAN1; ade2-1; his3-11; leu2-3-112; ura3-1; trp1-1* |
| 11,137, 11,252, 11,253 | a | *gcn5::HIS3; rDNA::ADE2-CAN1; ade2-1; his3-11; leu2-3-112; ura3-1; trp1-1* clones 1-3 |
| 11,144, 11,145, 11,146 | a | *fob1::hphNT1; rDNA::ADE2-CAN1; ade2-1; his3-11; leu2-3-112; ura3-1; trp1-1* clones 1-3 |
| 11,150, 11,151, 11,152 | a | *sgf73::hphNT1; rDNA::ADE2-CAN1; ade2-1; his3-11; leu2-3-112; ura3-1; trp1-1* clones 1-3 |
| 11,153, 11,154, 11,155 | a | *sir2::hphNT1; rDNA::ADE2-CAN1; ade2-1; his3-11; leu2-3-112; ura3-1; trp1-1* clones 1-3 |
| 10,998, 10,999, 11,000 | a | *sac3::natNT2; gcn5::hphNT1; pPCM14 (224 tetO-REC-URA3-CEN-REC-LEU2); spc42::SPC42-CFP:kanMX4; leu2::TETR-GFP:LEU2; his3::P$_{GAL}$-REC:HIS3; trp1:: GAL4-EBD:TRP1; ade2-101;* clones 1-3 |

## Fluorescent microscopy

For microscopy, cells were resuspended in non-fluorescent medium (SD–TRP) and either mounted on a glass cover slip (for still images or movies of max. 5 min duration) or immobilized on a 2% agar pad containing non-fluorescent medium.

To visualize, the accumulated plasmids cells were grown on SD–URA to maintain the centromere on the plasmid, then shifted to SD -LEU containing β-Estradiol (1 μM final concentration; Sigma-Aldrich, St. Louis, MO) to induce excision of the centromere (*rec-CEN4-URA3-rec*) and to select for the presence of the plasmid. After 16–18 hr, a Deltavision microscope (Applied Precision) equipped with a CCD HQ2 camera (Roper), 250W Xenon laps, Softworx software (Applied Precision), and a temperature chamber set to 30°C was used to visualize the plasmids and the depicted proteins tagged with GFP. Time lapse movies of 5 min, intervals for 1h taking 20 Z-stacks of 0.4-μm spacing, were recorded with a 100×/1.4 NA objective and 2x2 binning. To determine the total NPC fluorescence, integrated density was measured of the nucleus and background in sum projections using Image J (National Institutes of Health). After subtraction of the background, the fluorescence was normalized by the median fluorescence of cells without plasmids. To determine fluorescence intensity, the mean gray value of the NPC area adjacent to the accumulated circles (Ic) and the residual NPC area (Ir) was measured and normalized by the median intensity of Ir. For the percentage fluorescence segregated to the mother cell, the integrated density of the nucleus in the mother and the bud was measured after nuclear division, and after subtraction of the background the fluorescence in the mother was divided by the total fluorescence. The nuclear area depicts the 2D area of the nucleus in a sum projection in the mother and daughter of telophase cells. The mean intensity is the total intensity divided by the nuclear area. Box plots represent the distribution of the measurements in single cells whereas whiskers reach from min to max, the box covers 50% of the data and the line depicts the median.

To investigate plasmid propagation, the cells were kept on SD–URA medium and diluted in YPD medium containing estradiol only 3 hr before imaging. The cells were washed once using 1× PBS buffer and then resuspended in non-fluorescent medium. Images were obtained using an Olympus BX50 microscope, equipped with a piezo motor, a monochromator light source, a CCD camera (Andor), and the TillVision software (TillPhotonics). Images were acquired with a 100x / 1.4 NA objective, 2 × 2 binning, and stacks of 20 focal slices. Maximal intensity projections were used to determine plasmid propagation (described below 'plasmid retention assay').

To determine individual plasmid localization, accumulated plasmid co-localization with Gcn5 and Sgf73, nuclear geometry, and anaphase duration; a Deltavision microscope was used (described above). Prior to analysis, the images were deconvolved. For plasmid localization, images were taken with an Olympus 100×/1.4 NA objective and 1 × 1 binning. 15 focal sections with 0.3-μm spacing were acquired. The focal plane in which the plasmid was the brightest was used to determine whether the plasmid was at the nuclear rim or at a clearly resolvable distance from the nuclear periphery. Nuclear geometry was analyzed using an Olympus 100×/1.4 NA objective, 1 × 1 binning, and a 1.6× auxiliary magnification. Stacks of 15 sections and 0.2-μm spacing were obtained. Nuclear length and width was

measured as illustrated in *Figure 4B* using ImageJ, and average nuclear width was calculated for different categories of nuclear length. Anaphase duration was determined using an Olympus 60×/1.42 NA objective and 1 × 1 binning. 30 min movies were taken with 1 min between each stack of 8 sections and 0.6-μm spacing. Early anaphase was measured form the first time point the nucleus crossed the bud neck until the nucleus contracts as shown in the cartoon and representative pictures in *Figure 4A*.

To investigate the enrichment of NPCs close to non-centromeric plasmids, a Core Deltavision containing a CCD HQ camera (Roper) and solid state LEDs was used. An Olympus 100×/1.4 NA objective, 1 × 1 binning was used to create stacks of 15 focal sections and 0.2-μm spacing. Using ImageJ fluorescence intensity was measured along the nuclear rim. Nup82-3sfGFP intensity was normalized by its average intensity and TetR-mCherry intensity was normalized by its highest intensity in each cell. Afterwards, all obtained traces were aligned by the peak of TetR-mCherry (plasmid) intensity.

To determine the speed of the non-centromeric plasmids, cells were imaged 3h after centromere excision using the same Deltavision with LEDs. First a Z-stack of 16 sections and 0.3-μm spacing was recorded in CFP and YFP to visualize both the plasmid and the SPBs in order to select for the non-centromeric plasmids. Then the plasmid was followed in a time lapse movie, taking the same stacks but only in GFP every 3 s for 3 min. The plasmids were tracked using Imaris (Bitplane) to determine their three dimensional coordinates. The coordinates were then used to determine the distance covered by the plasmid for each time point, which was deviated by three to determine the distance per second. All speed values were averaged for every plasmid, then averaged over all non-centromeric plasmids recorded and plotted in the graph.

## FLIP and FRAP experiments

Both FLIP and FRAP experiments were essentially done as described in *Boettcher et al. 2012*. Briefly, to measure the dynamics of NPCs in cells with and without accumulated plasmids, cells were grown in SD -URA, restreaked on SD–LEU containing β-estradiol (1 μM final concentration; Sigma) 16–18 hr prior to imaging, and then immobilized on a 2% agar pad containing non-fluorescent medium. A Zeiss LSM 780 laser confocal microscope (Zeiss Microimaging) with a 63×/1.4 NA objective and a multiarray 32PMT GaAsP detector controlled by the ZEN 2011 software (Carl Zeiss) was used for imaging. Bleaching was performed using a 488 nm argon laser at 2.6% laser intensity and 45% output. Repeated photobleaching was performed in the depicted area with 9 s intervals for 30 time points. Afterwards the fluorescence intensity of Nup170-GFP was measured in the vicinity of the circles ($I_c$) and an area of the same size equidistant from the bleaching region ($I_o$) using ImageJ. Both intensities were set to 100% prior to the first bleaching. Fluorescence decay was plotted for each cell using Prism (GraphPad Software) and $t_{70}$ was defined as the time after which 70% of the initial intensity was still recorded. Finally, the average $t_{70}$ of several cells was calculated for both $I_c$ and $I_o$ and shown in the graph. The same settings were used to investigate the fluorescence decay in the whole nucleus, whereas the whole nuclear area was measured and set to 100% prior to bleaching. The mean and standard deviation of individual cells was plotted in the graph.

To determine the strength of the nuclear diffusion barrier, a Zeiss LSM 510 microscope was used. Early anaphase cells were selected and photobleaching was applied on the depicted areas. To analyze the FLIP experiments, the total integrated fluorescent density in both the mother and daughter part of the nucleus was measured. After subtracting the average fluorescent loss determined by 5 neighboring cells, the fluorescent signal of the mother and daughter part was set to 100% at the beginning of the experiment. All experiments were pooled and analyzed using Prism to fit a one-phase decay curve. The barrier index was defined as the ratio of the times needed to lose 40% of the initial fluorescent signal in the bud over the mother compartment.

For the FRAP experiments, the whole bud compartment of early anaphase nuclei was bleached and fluorescent recovery was measured in a constant area located in the bud. After subtracting the average fluorescent loss by imaging based on three neighboring cells, the fluorescence after photobleaching was set to 0%. The fluorescence recovery was fitted to a one-phase association curve for each cell using Prism. $T_{15}$ was defined as the time to recover 15% fluorescence. The average of $t_{15}$ of all measured cells was calculated and is shown in the graph.

## Plasmid retention assay

Late anaphase or telophase cells containing one, two, or four plasmids that did not co-localize with the spindle pole body were analyzed. Propagation frequency (pf) was calculated as;

100-(xa+2y$\sqrt{}$b+4z$^4\sqrt{}$c)/(x+2y+4z), where a, b, and c being the percentages of cells retaining all plasmids in the mother cell and x, y, and z being the number of cells counted for one, two, or four plasmids, respectively.

Retention of pDS316 (*Sinclair and Guarente, 1997*) was analyzed using freshly transformed wt and mutant cells. Cells were grown on SD–ADE plates and then transferred to plates containing low amounts of adenine 3h before micromanipulations. Using microdissection techniques, mother cells were separated from their daughter cells. After 3–5 days, both the mother and the daughter cells formed a colony, whereas the appearance of white sectors depicted whether the original cell contained pDS316.

## Southern blot analysis of aged cells

Old cells were purified following the protocol of the mother enrichment program (*Lindstrom and Gottschling, 2009*). Briefly, $2.5 \times 10^8$ cells were labeled with Sulfo-NHS-LC-Biotin (Pierce) and recovered for 2hr at 30°C prior to the addition of β-Estradiol (1 µM final concentration). After either 26 hr and 27.5 hr or 34 hr and 35.5 hr (for wt and mutant cells) cells were washed, diluted in PBS, and incubated with 25 µl streptavidin-coated magnetic beads (MicroMACS, Miltenyi Biotec) for 30 min at 4°C before purification using LS MACS columns (Miltenyi Biotec). Cells were eluted using 1× PBS containing 2 mM EDTA, pelleted, and split into two fractions: 1) cells were incubated in 20 µg/ml calcofluor white and bud scars were counted using fluorescence microscopy; 2) cells were lysed and DNA was purified using standard methods. DNA content was measured performing qPCR amplifying *ACT1* as follows: genomic DNA from wt non-aged cells was used to generate a dilution curve. Simultaneously, three qPCRs for two different dilutions were run for each genomic DNA sample derived from the aged wt, *gcn5Δ* and *sgf73Δ* mutant cells. The average of the six qPCR values obtained for each strain was used to calculate the volume loaded on a 0.6% agarose gel containing ethidium bromide. The gel was run in TBE containing ethidium bromide for 24 hr at 50 V at 4°C. The gel was blotted to transfer membrane (magnacharge nylon, GE Water&Process Tech.) using standard protocols. Membranes were hybridized with a $^{32}$P-labeled probe specific to the rDNA as described (*Lindstrom et al., 2011*) and visualized using a Typhoon phosphoimager (GE healthcare).

## ERC formation assay

Cells containing one copy of the *ADE2* gene inserted in the rDNA array were grown over night in SD -ADE medium to prevent premature loss of the marker. After dilution the cells were grown for 4–6 hr, diluted again and plated on YPD plates lacking extra Adenine. Colonies were grown for 3 days at 30°C and then shifted to 4°C for two days before analysis. Half-sectors were counted and the total amount of colonies was assayed using an automated colony counter developed by Sandra Guetg (Barral lab, unpublished).

## ChIP assay

ChIP analysis was essentially done as described (*Chymkowitch et al., 2012*): Cells were fixed with 1% (wt/vol) formaldehyde for 30 min and quenched for 5 min using glycine (125 mM final concentration). Cells were washed with Tris-buffered saline, resuspended in lysis buffer, and lysed in a cooled freezer mill (Spex) for 6 cycles (2 min with 2 min breaks each at 12 cps). Afterwards, the samples were sonicated twice for 15 min by altering cycles of 30 s pulses followed by 30 s cool down periods using a Bioruptor (Diagenode). After centrifugation, supernatants were immunoprecipitated overnight at 4°C on anti-GFP beads (Chromotek). Washing steps and elution were performed as described (*Chymkowitch et al., 2012*). After RNase and proteinase K treatment, DNA fragments were purified using QIAquick PCR purification kit (Qiagen) and were analyzed by qPCR using a rotorgene qPCR system (Qiagen). Primer sequences were as follows:

ChrV: GAACAAGGTTACAAATCC and CCGATTTGTGAGATTCTTCCT

ARS: AGTAAAGATTTCGTGTTCATGCAG and ATAGTGAAGGAGCATGTTCGG non-coding sequence (NCS1): GATACCTGAGTATTCCCACAG and CACACCGCATATGGTGCACTC non-coding sequence (NCS2): CCGCTTACCGGATACCTGTC and ATCCTGTTACCAGTGGCTGC

Fold enrichment was calculated as IP/Input of protein X normalized by the same ratio measured in wt cells, where no protein was tagged with GFP to correct for unspecific binding.

## Microfluidic experiments

Pore segregation during ageing was investigated using the microfluidics dissection platform (*Lee et al., 2012*; *Huberts et al., 2013*) with constant flow of synthetic complete medium at 1 µl/min and an

Eclipse Ti fluorescent microscope (Nikon Instruments) controlled by Micro-Manager 1.4 software (µManager). Images were acquired with a Plan Apo lambda 60×/1.4 NA objective and a Orca R2 CCD camera (Hamamatsu). Cell growth and budding events were monitored by capturing images using transmitted light every 20 min. Fluorescent images were taken after 24 hr and 36 hr for short-lived strains (e.g. *sir2Δ* cells) and after 48 hr and 64 hr for normal or long-lived strains using a mercury lamp as a light source (Intensilight, Nikon Instruments). Stacks of 14 slices covering a range of 7 µm were captured every 15 min for 2 hr, to follow the nuclear division. Nuclei of cells of different ages were analyzed using sum projections of the fluorescence images from both time points (48 hr and 64 hr). Total fluorescence (integrated density) was measured in the mother and daughter cell after nuclear division using ImageJ. After background subtraction, the values were grouped by the indicated age categories of the mother cells, normalized by the median of the daughter cells in the youngest age category, and plotted using Prism. Alternatively, nuclear fluorescence in the mother cells was divided by the total fluorescence (mother plus daughter nuclei) for each cell and plotted against the age of the mother cell to visualize the increase in asymmetry with age. This data were used to fit a one-phase association curve, which is shown in the graph. Additionally, 10 values were grouped age dependently and plotted showing the average age and percentage of fluorescence segregated to the mother cell as well as the standard deviations. For G1 cells, the total amount of NPCs (integrated density) was measured, plotted against their age, and an exponential curve was fitted using Prism. The values were then normalized by intersection of the curve with the Y-axis and set to 1. The dots on the graph were created from 10 values as described for the dividing cells. Similarly, the mean fluorescence intensity (integrated density divided by area) was plotted as well as the radius of the G1 nuclei, measured drawing a line through the nucleus at its broadest area using ImageJ, dividing it by two and plotting it against the age of the cell.

### Lifespan analysis

Pedigree analysis was done as described (*Shcheprova et al., 2008*): freshly streaked cells were restreaked on YPD plates and 2 hr later virgin daughter cells were placed to analyze their lifespan. Over-night the cells were kept at 4°C (max. 10-12 hr). Micromanipulation was performed using a Zeiss Axioscope 40 microdissection microscope (Zeiss). Lifespan curves were analyzed using Prism.

### Statistics

One-way ANOVA followed by Dunnett's multiple comparison or an unpaired two-tailed t-test was used to test for significance except the lifespan curves, which were compared using Log-rank (Mantel–Cox) tests and the ChIP experiment, for which a paired t-test (within the same experiment) was used.

## Acknowledgements

We thank K Weis, U Kutay, T Kruitwagen, F Caudron, C Weirich, and R Kroschewski for critical reading of the manuscript, the whole Barral and Kroschewski groups for helpful discussions, S S Lee for the SU-8 master mold and technical support, P Chymkowitch, J Enserink, E Fabre, and the light microscopy center (ScopeM) for technical help. This work was supported by the ETH Zurich, an ERC and a System-X grant.

## Additional information

### Funding

| Funder | Author |
| --- | --- |
| European Research Council | Annina Denoth-Lippuner, Marek Konrad Krzyzanowski, Catherine Stober, Yves Barral |

The funder had no role in study design, data collection and interpretation, or the decision to submit the work for publication.

### Author contributions

AD-L, Acquisition of data, Analysis and interpretation of data, Drafting or revising the article; MKK, Acquisition of data, Drafting or revising the article; CS, Acquisition of data, Analysis and interpretation of data; YB, Conception and design, Drafting and revising the article

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
