## [Decision Letter]

Thank you for sending your work entitled "Role of SAGA in the asymmetric segregation of DNA circles during yeast ageing" for consideration at *eLife*. Your article has been favorably evaluated by Randy Schekman (Senior editor), a guest Reviewing editor, and 2 reviewers.

The Reviewing editor and the reviewers discussed their comments before we reached this decision, and the Reviewing editor has assembled the following comments to help you prepare a revised submission.

The paper is interesting and worthy of publication. The reviewers agree with the conclusions that, as yeast cells age, ERCs interact with the NPC through a mechanism that requires SAGA and that this alters the behavior (and potentially the function) of NPCs. However the reviewers also agree that the authors have overstated the mechanistic contribution of SAGA in the linkage of circular DNAs to the NPC. This latter conclusion is unsubstantiated without quite a bit more experimental evidence. To conclude that SAGA is really mediating the interaction with the NPC, the authors need to better define this mechanism. Does SAGA bind equally throughout the plasmid after eviction of the CEN? How does the E173A mutation impact this binding? How different is the acetylation of the histones associated with centromeric vs the non-centromeric plasmid? Therefore, it is recommended that the authors may proceed if they remove these claims. Nevertheless, even to make the case for the SAGA linkage at all, there are couple of things that must be addressed, some with simple straightforward experiments: 1) In the plasmid retention assay, perform the double mutant *gcn5∆ sac3∆* (or *ada5∆sac3∆*
*–* better) measurements to properly test that there is no additive effect in eliminating TREX and SAGA.

2) The quantitative data for NPC cap formation in a *bud6∆* mutant should be shown in Figure 2.

3) In Figure 11, another interpretation of the data is that *GCN5* and *SPT20* are required for the RLS extension caused by deleting *FOB1*, possibly because the *gcn5∆* mutant already has reduced ERC accumulation (as shown in Figure 3).

---

## [Author Response]

*The paper is interesting and worthy of publication. The reviewers agree with the conclusions that, as yeast cells age, ERCs interact with the NPC through a mechanism that requires SAGA and that this alters the behavior (and potentially the function) of NPCs. However the reviewers also agree that the authors have overstated the mechanistic contribution of SAGA in the linkage of circular DNAs to the NPC*.

We agree with the reviewers’ comment that we don’t know precisely how SAGA promotes the interaction of DNA circles with NPCs. SAGA might be part of a physical linker or might act by modifying chromatin or pore components to promote interaction between the circles and the NPCs. We have edited the manuscript to make sure that there is no confusion about this point and added a paragraph in the Discussion section to address the different possibilities of how SAGA would promote attachment of DNA circles to NPCs.

*1) In the plasmid retention assay, perform the double mutant gcn5∆ sac3∆ (or ada5∆sac3∆ – better) measurements to properly test that there is no additive effect in eliminating TREX and SAGA*.

To address the reviewers’ suggestion we created both double mutant strains. Unfortunately, in our hands the *ada5Δ sac3Δ* double mutant cells were very sick. Therefore, we analyzed the *gcn5Δ sac3Δ* double mutant cells. Simultaneously deleting *GCN5* and *SAC3* does not lead to additive effects compared to the single mutants, supporting the hypothesis that SAGA and TREX-2 function together in the same pathway for what concerns DNA circle retention in the mother cell. These data were added to the manuscript in Figure 5.

*2) The quantitative data for NPC cap formation in a bud6∆ mutant should be shown in*
Figure 2.

As requested by the reviewers, we added these results to Figure 2. Similar to wt cells, the NPC signal is twice as strong around the plasmids compared to the residual nuclear area.

*3) In*
Figure 11*, another interpretation of the data is that GCN5 and SPT20 are required for the RLS extension caused by deleting FOB1, possibly because the gcn5∆ mutant already has reduced ERC accumulation (as shown in*
Figure 3*)*.

We agree with the reviewers that this is a possibility; however this is unlikely to be due to a reduced rate of ERC formation. Indeed, we are now adding data on the effect of SAGA inactivation on recombination in the rDNA. As we show now, the *gcn5∆* mutant cells do not show a reduced frequency of rDNA excision; if anything it is rather slightly increased. Thus, the only thing that we can conclude from these data is that in both the *gcn5Δ* single and *gcn5Δ fob1Δ* double mutant cells ERC accumulation is no longer the factor limiting longevity.

Our new data also permit to extend the discussion opened by a recent study by [45], who suggest that the effect of Sgf73 on ageing is partially due to a decrease in ERC formation. Using the same test as for the *gcn5∆*, we confirmed that indeed, cells lacking Sgf73 show a reduction (2 folds) in ERC formation, similar to the cells lacking Fob1. However, the *sgf73∆* mutant cells show a much more prolonged lifespan compared to the *fob1∆* mutant cells. Thus, the effect of the *sgf73∆* mutation on longevity probably involved a combination of effects, such as a reduction of ERC formation and the lack of retention of the residual ERCs. This interpretation is still fully supported by the fact that restoring SAGA linkage to NPCs in these cells restores ageing.